# Artificial Intelligence-Enhanced Precision Medicine Reveals Prognostic Impact of TGF-Beta Pathway Alterations in FOLFOX-Treated Early-Onset Colorectal Cancer Among Disproportionately Affected Populations

**DOI:** 10.3390/ijms26189067

**Published:** 2025-09-17

**Authors:** Fernando C. Diaz, Brigette Waldrup, Francisco G. Carranza, Sophia Manjarrez, Enrique Velazquez-Villarreal

**Affiliations:** 1Center for Cancer Research, National Cancer Institute, Bethesda, MD 20814, USA; 2City of Hope, Beckman Research Institute, Department of Integrative Translational Sciences, Duarte, CA 91010, USA; 3City of Hope Comprehensive Cancer Center, Duarte, CA 91010, USA

**Keywords:** colorectal cancer, precision medicine, cancer genomics, TGF-beta pathway, artificial intelligence, LLM, chemotherapy, cancer disparities, cancer diagnosis, cancer treatment

## Abstract

Early-onset colorectal cancer (EOCRC; <50 years) incidence is increasing most rapidly among Hispanic/Latino (H/L) populations. While the transforming growth factor–beta (TGF-β) pathway influences colorectal cancer (CRC) progression, its prognostic role in FOLFOX-treated EOCRC, particularly in H/L patients, is unclear. We analyzed 2515 CRC cases (H/L = 266; NHW = 2249) stratified by ancestry, age at onset, and FOLFOX treatment using Fisher’s exact, chi-square, and Kaplan–Meier analyses. We then applied AI-HOPE and AI-HOPE-TGFβ, conversational artificial intelligence (AI) platforms that integrate clinical, genomic, and treatment data, to perform complex, natural language-driven queries requiring multi-parameter integration. TGF-β pathway alterations occurred in 28–39% of H/L and 23–31% of NHW patients, with SMAD4 being the predominant driver. BMPR1A mutations were enriched in FOLFOX-treated EO H/L patients (5.5% vs. 1.1% EO NHW; *p* = 0.0272), while late-onset NHW non-FOLFOX cases had higher SMAD2/TGFBR2 mutation rates. In FOLFOX-treated EO H/L patients, TGF-β pathway alterations predicted poorer survival (*p* = 0.029); no survival impact was seen in other groups. SMAD4 mutations were less frequent in EO H/L than in EO NHW receiving FOLFOX (2.74% vs. 13.87%; *p* = 0.013). TGF-β pathway alterations may serve as ancestry- and treatment-specific biomarkers of poor prognosis in FOLFOX-treated EO H/L CRC. AI-enabled integration accelerated biomarker discovery, supporting precision medicine.

## 1. Introduction

Colorectal cancer (CRC) has traditionally been considered a disease of older adults, with incidence rates historically declining or stabilizing in high-income countries due to advances in prevention and widespread access to early screening programs [1]. Despite this progress, CRC remains the third most commonly diagnosed cancer and ranks as the third leading cause of cancer-related deaths in men and the fourth in women [2]. In contrast to the overall decline, the incidence of early-onset colorectal cancer (EOCRC)—defined as diagnosis before age 50—has risen sharply over the past two decades and is projected to become the leading cause of cancer-related deaths among individuals aged 20 to 49 in the United States by 2030 [3,4,5,6,7,8,9]. This increase is particularly pronounced among Hispanic/Latino (H/L) populations, who have experienced some of the steepest rises in EOCRC incidence and mortality in recent years [2,10,11,12]. Representing 14% of the U.S. GDP (~$3.2 trillion) and 18% of the national workforce [13,14], the H/L population’s growing cancer burden underscores a pressing public health and economic challenge. Understanding the genomic and molecular drivers of EOCRC in these disproportionately affected groups is essential for guiding prevention, drug development, and personalized treatment strategies.

Although emerging research has identified important molecular differences between EOCRC and late-onset colorectal cancer (LOCRC), findings remain inconsistent. Discrepancies in reported tumor mutation burden, microsatellite instability (MSI) status, and PD-L1 expression have been documented [15,16,17]. Nonetheless, several studies have highlighted unique biomarkers in EOCRC—including LINE-1 hypomethylation and distinct mutational profiles involving genes such as SMAD4, TP53, APC, and KRAS—suggesting a potentially different biological trajectory compared to LOCRC [15,16,17,18,19]. Most prior genomic studies, however, have been conducted in predominantly non-Hispanic White (NHW) populations, leaving critical gaps in our understanding of EOCRC molecular features within H/L and other underserved groups [20]. Our recent work has begun addressing this gap by characterizing the mutational landscape of key oncogenic pathways, including WNT, MAPK, JAK/STAT, PI3K, and TGF-beta, in H/L EOCRC patients [21,22,23].

The TGF-beta pathway plays a pivotal role in regulating cell differentiation, growth, apoptosis, and adhesion [24,25]. In cancer, TGF-beta signaling exhibits a context-dependent dual role: functioning as a tumor suppressor in early stages while promoting epithelial-to-mesenchymal transition (EMT) and contributing to an immunosuppressive tumor microenvironment in advanced disease [26,27]. SMAD4 alterations have been implicated in EMT progression and poor outcomes in metastatic CRC [27], while other TGF-beta pathway genes—including BMP7, TGFBR2, and ACVR1B—have been linked to CRC pathogenesis and prognosis [21,28,29]. Importantly, our previous findings revealed a higher prevalence of BMP7 alterations in H/L EOCRC patients, with improved outcomes observed in those lacking such alterations [21].

For metastatic microsatellite-stable (MSS) CRC without actionable mutations and proficient mismatch repair (pMMR), the American Society of Clinical Oncology (ASCO) recommends the FOLFOX regimen—comprising folinic acid, fluorouracil (5-FU), and oxaliplatin—as the standard first-line treatment [30,31]. However, EOCRC patients treated with FOLFOX appear to experience poorer overall survival and higher treatment-related toxicity compared to LOCRC patients [32]. The extent to which alterations in the TGF-beta pathway influence FOLFOX responsiveness in EOCRC, particularly in disproportionately affected populations, remains largely unexplored.

Recent advances in artificial intelligence (AI) have opened new avenues for addressing these knowledge gaps. Conversational AI agents, such as our AI-HOPE [33] platform, enable dynamic integration of clinical, genomic, and molecular data to conduct complex precision medicine analyses. The specialized AI-HOPE-TGFbeta [34] module is designed to interrogate TGF-beta pathway alterations in CRC, rapidly generating pathway-specific queries, identifying clinically relevant mutation patterns, and correlating these with treatment outcomes. By enabling iterative, natural language-driven exploration of large-scale datasets, AI agents overcome traditional bioinformatics bottlenecks and facilitate hypothesis generation that is both comprehensive and adaptable. Incorporating AI-HOPE-TGFbeta into this study allows us to systematically evaluate the relationship between TGF-beta pathway alterations and FOLFOX treatment outcomes in EOCRC, with a focus on disproportionately affected populations. This approach not only advances mechanistic understanding but also exemplifies how AI-driven tools can accelerate the translation of genomic insights into actionable clinical strategies.

## 2. Results

### 2.1. Baseline Clinical and Demographic Profiles of H/L and NHW CRC Cohorts

The clinical and demographic characteristics of the Hispanic/Latino (H/L) and Non-Hispanic White (NHW) colorectal cancer (CRC) cohorts—stratified by age at onset, FOLFOX treatment status, tumor type, sex, disease stage, microsatellite instability (MSI) status, and detailed ethnicity annotations—are presented in Table 1.

Within the H/L cohort (*n* = 266), EOCRC patients treated with FOLFOX accounted for 27.4% of cases, compared to 16.7% in the NHW cohort (*n* = 2249). In contrast, LOCRC cases treated with FOLFOX were more common among NHW patients (40.9%) than H/L patients (34.2%). A higher proportion of EOCRC cases without FOLFOX treatment was observed in the H/L group (19.5%) compared to NHW patients (13.4%).

Regarding tumor type, colon adenocarcinoma was the most frequent diagnosis in both groups (61.7% H/L; 59.0% NHW), followed by rectal adenocarcinoma (24.1% H/L; 28.7% NHW) and combined colorectal adenocarcinoma (14.3% H/L; 12.2% NHW). Sex distribution was comparable between cohorts, with a slight predominance of males in both groups. All samples represented primary tumors.

Stage at diagnosis showed similar patterns across groups, with most patients diagnosed at stages 1–3 (58.6% H/L; 55.0% NHW), while stage 4 disease was slightly less frequent in H/L patients (40.6%) than in NHW patients (44.7%). MSI stability predominated in both groups, though MSI-stable tumors were less frequent in H/L patients (75.2%) compared to NHW patients (86.3%), and a higher proportion of MSI status was missing in the H/L cohort (13.2% vs. 1.9%).

As expected, ethnicity sub-classifications confirmed complete separation between the groups: all NHW patients were recorded as non-Spanish/non-Hispanic, while H/L patients were most frequently annotated as “Spanish NOS; Hispanic NOS; Latino NOS” (86.5%), followed by “Mexican (includes Chicano)” (11.3%), with smaller representation from other H/L categories.

### 2.2. Genomic Comparisons Across Age Groups and Ancestral Backgrounds

#### 2.2.1. H/L Patients by Age and Treatment Status

Clinical and genomic features of Hispanic/Latino (H/L) colorectal cancer patients, stratified by age at diagnosis and FOLFOX treatment status, are summarized in Table 2a. Among early-onset cases, FOLFOX-treated patients had a slightly higher median diagnosis age (42 years; IQR 36–47) than those not treated (40 years; IQR 34–43), a difference that approached but did not reach statistical significance (*p* = 0.0541). Among late-onset patients, median diagnosis age was significantly younger in the FOLFOX-treated group (59 years; IQR 54–66) compared with the non-treated group (62 years; IQR 56–70; *p* = 0.0487). Median mutation counts were comparable across treatment groups for both age categories. TMB showed a non-significant trend toward higher values in early-onset non-FOLFOX cases, whereas in late-onset patients, TMB was significantly greater in non-FOLFOX cases (6.9; IQR 5.6–9.0) than in FOLFOX-treated cases (6.1; IQR 4.9–7.8; *p* = 0.0439). Fraction of genome altered (FGA) did not vary significantly between groups, and BMPR1A mutations were rare across all categories without treatment-related differences.

#### 2.2.2. NHW Patients by Age and Treatment Status

Findings for NHW patients are outlined in Table 2b. Among early-onset cases, the median diagnosis age did not differ between FOLFOX-treated (43 years; IQR 37–48) and non-treated patients (44 years; IQR 38–47; *p* = 0.5646). In contrast, late-onset FOLFOX-treated patients were significantly younger at diagnosis than non-treated patients (63 vs. 66 years; *p* = 4.15 × 10^−7^). Mutation counts were similar between early-onset treatment groups but significantly higher in late-onset non-FOLFOX patients (8; IQR 6–12) compared to treated patients (7; IQR 5–9; *p* = 1.22 × 10^−5^). TMB patterns mirrored these findings, with higher values in late-onset non-FOLFOX patients (6.6 vs. 6.1; *p* = 0.000285), but no differences in early-onset patients. FGA values did not differ significantly between treatment groups. BMPR1A mutations were rare. SMAD2 mutations were significantly more frequent in late-onset non-FOLFOX patients (6.6%) than in treated patients (3.8%; *p* = 0.0173). SMAD3 mutations were slightly higher in late-onset non-FOLFOX cases (5.2%) compared to treated cases (3.2%), though not statistically significant (*p* = 0.0558). TGFBR2 mutations were significantly more common in late-onset non-FOLFOX patients (7.0%) than in FOLFOX-treated patients (4.1%; *p* = 0.0158).

#### 2.2.3. Ethnic Comparisons in Early-Onset Disease

Table 2c compares early-onset H/L and NHW patients by treatment status. Among FOLFOX-treated early-onset patients, median diagnosis age was slightly lower in H/L patients (42 years; IQR 36–47) than NHW patients (43 years; IQR 37–48), but this was not statistically significant (*p* = 0.0847). In the non-FOLFOX group, H/L patients were diagnosed significantly earlier (40 vs. 44 years; *p* = 0.00060). Mutation counts and TMB were similar between ethnic groups in both treatment categories, although TMB trended higher in FOLFOX-treated H/L patients (6.3 vs. 5.7; *p* = 0.0581). FGA values were comparable. Notably, BMPR1A mutations were significantly more frequent in FOLFOX-treated H/L patients (5.5%) than in treated NHW patients (1.1%; *p* = 0.0272), while prevalence in the non-FOLFOX group was similar between ethnicities.

#### 2.2.4. H/L Patients: Early-Onset vs. Late-Onset

Early-onset and late-onset H/L patients are compared within treatment categories in Table 2d. As expected, median diagnosis age was markedly lower in early-onset patients for both the FOLFOX-treated (42 vs. 59 years; *p* = 2.20 × 10^−16^) and non-FOLFOX groups (40 vs. 62 years; *p* = 2.20 × 10^−16^). In the FOLFOX-treated cohort, late-onset patients had slightly higher mutation counts than early-onset patients (8 vs. 7; *p* = 0.0172), while no significant differences were seen in the non-FOLFOX cohort. TMB was similar between age groups in FOLFOX-treated patients but was significantly higher in late-onset non-FOLFOX patients (6.9 vs. 5.5; *p* = 0.0328). FGA values were similar across age groups. BMPR1A mutations occurred exclusively in early-onset patients (5.5% in FOLFOX-treated, 3.8% in non-treated), with no cases in late-onset disease.

### 2.3. TGF-Beta Pathway Alterations by Age, Ancestry, and Treatment Status

The prevalence of TGF-beta pathway alterations was examined across H/L and NHW CRC cohorts, stratified by age at diagnosis and FOLFOX treatment status (Table 3a–d).

Within the H/L cohort, alteration frequencies were similar between treatment groups in both early- and late-onset disease. In early-onset H/L patients, TGF-beta alterations were observed in 28.8% of FOLFOX-treated cases and 36.5% of non-treated cases (*p* = 0.4693). In late-onset patients, alterations occurred in 33.0% of treated cases and 38.0% of non-treated cases (*p* = 0.6777).

Within the NHW cohort, early-onset patients showed no significant difference in alteration prevalence between FOLFOX-treated (24.5%) and non-treated (26.2%) groups (*p* = 0.6929). In contrast, among late-onset NHW patients, alterations were significantly less common in the FOLFOX-treated group (23.4%) than in the non-FOLFOX group (31.4%; *p* = 0.00051).

Between-ancestry comparisons in early-onset disease showed no statistically significant differences in TGF-beta alteration prevalence. Among FOLFOX-treated early-onset patients, alterations were detected in 28.8% of H/L cases and 24.5% of NHW cases (*p* = 0.5387). In non-FOLFOX patients, prevalence was 36.5% in H/L and 26.2% in NHW (*p* = 0.1684).

Between-ancestry comparisons in late-onset disease revealed that, among FOLFOX-treated patients, H/L cases had a higher alteration prevalence (33.0%) than NHW cases (23.4%), a difference that approached statistical significance (*p* = 0.0569). In non-FOLFOX late-onset patients, alterations were present in 38.0% of H/L cases and 31.4% of NHW cases (*p* = 0.4186).

Across all stratifications, the majority of patients in both ethnic groups and age categories did not harbor TGF-beta pathway alterations. However, the higher prevalence observed in late-onset H/L patients receiving FOLFOX, as well as the significantly lower prevalence among late-onset NHW patients treated with FOLFOX, suggests potential ancestry- and treatment-specific differences in TGF-beta pathway involvement that may warrant further investigation.

### 2.4. Frequencies of Gene Alterations in the TGF-Beta Pathway

Across all stratifications by age of onset, ancestry, and FOLFOX treatment status, SMAD4 consistently emerged as the most frequently altered TGF-beta pathway gene in CRC patients.

In H/L early-onset patients (Appendix A), SMAD4 mutations occurred in 12.3% of FOLFOX-treated and 13.5% of non-FOLFOX cases. Other recurrent alterations included BMPR1A (5.5% treated; 3.8% untreated), SMAD2 (5.5%; 5.8%), SMAD3 (4.1%; 5.8%), TGFBR2 (4.1%; 11.5%), and TGFBR1 (2.7%; 3.8%). No significant differences were observed by treatment status, although TGFBR2 mutations were nearly threefold higher in the untreated group.

In H/L late-onset patients (Appendix A), SMAD4 remained the most altered gene (17.6% treated; 18.0% untreated), followed by TGFBR2 (5.5%; 6.0%), SMAD3 (4.4%; 8.0%), and SMAD2 (3.3%; 8.0%). BMPR1A mutations were absent in both treatment groups. No statistically significant differences were found between treated and untreated cases.

In NHW early-onset patients (Appendix A), SMAD4 mutations occurred in 13.9% of treated and 12.3% of untreated cases, followed by SMAD2 (5.1%; 5.3%), TGFBR2 (2.9%; 5.3%), BMPR1A (1.1%; 3.3%), SMAD3 (3.7%; 4.0%), and TGFBR1 (2.1%; 3.0%). No significant differences were observed, though BMPR1A mutations were more than twice as common in the untreated group (*p* = 0.0558).

In NHW late-onset patients (Appendix A), SMAD4 mutations were detected in 13.6% of treated and 15.9% of untreated cases. SMAD2 (3.8% vs. 6.6%; *p* = 0.0173) and TGFBR2 (4.1% vs. 7.0%; *p* = 0.0158) mutations were significantly more frequent in non-FOLFOX cases, with SMAD3 trending higher (3.2% vs. 5.2%; *p* = 0.0558).

When comparing early- vs. late-onset H/L patients treated with FOLFOX (Appendix A), BMPR1A mutations were significantly more common in early-onset cases (5.5% vs. 0.0%; *p* = 0.0375), while other genes showed no significant differences. In the non-FOLFOX group (Appendix A), mutation patterns were similar across age categories, with SMAD4 remaining most frequent (13.5% early-onset; 18.0% late-onset).

Among NHW patients, early- vs. late-onset comparisons revealed highly consistent mutation profiles regardless of treatment status. In FOLFOX-treated patients (Appendix A), SMAD4 was most frequent (13.9% vs. 13.6%) with no significant differences.

When comparing ancestries, in early-onset FOLFOX-treated patients (Appendix A), BMPR1A mutations were significantly more common in H/L than NHW (5.5% vs. 1.1%; *p* = 0.0272), while other genes showed similar frequencies. In the non-FOLFOX group (Appendix A), no significant ancestry-related differences were found, though TGFBR2 mutations were more than twice as frequent in H/L (11.5% vs. 5.3%; *p* = 0.1583).

In late-onset FOLFOX-treated patients (Appendix A), mutation distributions were similar between H/L and NHW, with SMAD4 being the most frequent (17.6% vs. 13.6%). In the non-FOLFOX group (Appendix A), SMAD4 again predominated (18.0% vs. 15.9%), with no significant differences between ancestries.

Overall, SMAD4 emerged as the predominant alteration across nearly all subgroups, while statistically significant differences were rare and primarily limited to BMPR1A (age- and ancestry-associated) and higher SMAD2/TGFBR2 mutation rates in late-onset NHW non-FOLFOX patients.

### 2.5. Mutational Landscape of the TGF-Beta Pathway

#### 2.5.1. Early-Onset H/L CRC

Figure 1a depicts the TGF-beta pathway mutational profile in H/L with EOCRC (*n* = 113; Appendix A), integrating mutation type, tumor mutational burden (TMB), and FOLFOX treatment status. Overall, 38 cases (33.6%) harbored at least one pathway alteration. SMAD4 was the most frequently mutated gene (14%), predominantly featuring missense mutations (green), alongside truncating events such as frame shift deletions (light blue) and nonsense mutations (red). TGFBR2 ranked second (8%), enriched for frame shift deletions and multi-hit events (black), with fewer missense variants. SMAD2 (6%) and SMAD3 (5%) exhibited mixed missense and truncating alterations, while BMPR1A (5%) mutations were largely missense with occasional splice site variants (orange). TGFBR1 (4%) alterations were mainly multi-hit events. TMB distribution was generally low, with a small subset of hypermutated cases lacking clear gene-specific clustering. FOLFOX-treated (blue) and untreated (red) cases occurred across all mutation categories without distinct segregation.

#### 2.5.2. Late-Onset H/L CRC

In LOCRC H/L (*n* = 123; Figure 1b), 48 cases (39.0%) carried at least one TGF-beta pathway alteration. SMAD4 was again the most frequently altered (20%), with a predominance of missense variants, along with nonsense mutations, frame shift deletions, and multi-hit events. TGFBR2 and SMAD3 each accounted for 7% of cases, with TGFBR2 enriched for frame shift deletions and missense variants, and SMAD3 showing missense, nonsense, and occasional splice site changes. SMAD2 was mutated in 6% of tumors, with a combination of missense and frame shift variants. TGFBR1 was altered in 3% of cases, exclusively via missense mutations. BMPR1A mutations were absent in this cohort. TMB was generally low, with no clustering of hypermutated cases by specific genes. FOLFOX treatment distribution was balanced across mutation categories.

#### 2.5.3. Early-Onset NHW CRC

In EOCRC NHW (*n* = 607; Figure 1c), 171 cases (28.2%) harbored at least one TGF-beta pathway alteration. SMAD4 was most frequently mutated (15%), with a spectrum of missense, nonsense, frame shift, in-frame deletion, and multi-hit events. SMAD2 ranked second (6%), followed by TGFBR2 (4%) and SMAD3 (4%), each showing a combination of missense and truncating variants, with TGFBR2 particularly enriched for frame shift deletions. Lower-frequency alterations were observed in TGFBR1 (3%) and BMPR1A (2%). TMB distribution was predominantly low, with a subset of hypermutated tumors. FOLFOX treatment status showed no distinct clustering of mutation types. Compared to EO H/L CRC, the EO NHW cohort displayed a lower overall prevalence of TGF-beta pathway alterations, suggesting possible ancestry-related differences in pathway dysregulation.

#### 2.5.4. Late-Onset NHW CRC

In LOCRC NHW (*n* = 1409; Figure 1d), 419 cases (29.7%) carried at least one TGF-beta pathway mutation. SMAD4 remained the most altered gene (16%), followed by TGFBR2 (6%) and SMAD2 (6%), both showing mixed missense and truncating events, and SMAD3 (4%), primarily with missense changes. BMPR1A and TGFBR1 each occurred in 2% of tumors, with BMPR1A displaying a diverse mutation spectrum and TGFBR1 largely limited to missense variants. TMB patterns were similar to those in other groups, with mostly low values and sporadic hypermutated cases. FOLFOX treatment distribution again showed no clear separation by mutation profile.

Overall, across all age and ancestry groups, SMAD4 consistently emerged as the most frequently altered TGF-beta pathway gene, followed by variable contributions from TGFBR2, SMAD2, SMAD3, BMPR1A, and TGFBR1. Mutation spectra were dominated by missense changes, supplemented by truncating and splice site events, reflecting diverse mechanisms of pathway disruption.

### 2.6. Survival Impact of TGF-Beta Pathway Alterations Across Age, Ancestry, and Treatment Groups

We investigated the relationship between TGF-beta pathway alterations and overall survival in colorectal cancer, stratifying patients by age of onset, ancestry, and FOLFOX treatment status using Kaplan–Meier survival analysis. In EOCRC H/L patients who received FOLFOX, the presence of TGF-beta pathway alterations was linked to a significant reduction in overall survival (*p* = 0.029; Figure 2a). Patients without alterations maintained survival rates near 100% for most of the follow-up, whereas those with alterations showed a steeper decline during the first ~50 months, followed by a plateau. The broader confidence intervals for the altered group reflect the smaller sample size. These results indicate that TGF-beta pathway mutations may adversely affect survival outcomes in EO H/L patients undergoing FOLFOX therapy. Among EO H/L patients who did not receive FOLFOX, there was no significant survival difference between altered and non-altered groups (*p* = 0.55; Figure 2b). Both groups maintained high survival rates with closely overlapping confidence intervals. A slight divergence appeared at later time points, but this was not statistically meaningful, and interpretation was limited by the small number of altered cases. For late-onset H/L (LO H/L) patients treated with FOLFOX, survival outcomes were similar regardless of alteration status (*p* = 0.65; Figure 2c). Although curves displayed slight early separation, they converged later, with overlapping confidence intervals and comparable median survival times, suggesting no substantial effect of TGF-beta pathway status. A similar pattern was observed in LO H/L patients not treated with FOLFOX, where survival curves were almost identical between groups (*p* = 0.93; Figure 2d), with overlapping confidence intervals and no meaningful divergence throughout follow-up. In EOCRC NHW patients treated with FOLFOX, no statistically significant difference in survival was found (*p* = 0.099; Figure 2e). However, a gradual separation of curves suggested a potential trend toward poorer outcomes in the altered group, particularly between 20 and 60 months. Despite this, overlapping confidence intervals indicate the trend may not be statistically reliable. Lastly, for EO NHW patients not treated with FOLFOX, survival was similar between groups (*p* = 0.37; Figure 2f). Curves showed minimal early divergence that diminished over time, with broad overlapping confidence intervals underscoring the lack of a measurable effect. Taken together, these analyses demonstrate that TGF-beta pathway alterations were significantly associated with worse survival only in EO H/L patients receiving FOLFOX, while no statistically significant associations were found in any other age–ancestry–treatment subgroup (Appendix A).

### 2.7. AI-Enabled Data Interrogation and Pre-Statistical Insights

The AI-HOPE [33] and AI-HOPE-TGF-Beta [34] platforms were first deployed to conduct a targeted, post-analysis scan of the integrated CRC datasets, rapidly generating exploratory insights that guided subsequent statistical testing. Initial natural language-driven queries revealed several patterns of interest.

First, among H/L CRC patients treated with FOLFOX, preliminary AI analysis indicated a potential association between TGF-β pathway alterations and reduced overall survival, which was confirmed by subsequent Kaplan–Meier analysis. In this subgroup (*n* = 21 altered; *n* = 52 non-altered), patients with TGF-β pathway alterations demonstrated significantly worse survival compared with their non-altered counterparts (log-rank *p* = 0.029) (Appendix A). This is consistent with our standard overall survival analysis results presented in Section 2.6 (Figure 2a). The survival curves showed an early and sustained separation, with the altered group experiencing a steeper decline in survival probability during the first ~50 months, followed by a plateau, whereas the non-altered group maintained higher survival probabilities throughout the follow-up period. The non-overlapping trend of the survival curves and narrower confidence intervals for the non-altered group suggest that TGF-β pathway alterations may represent a prognostic biomarker of poorer outcome in EO H/L patients receiving FOLFOX.

Second, AI-driven subgroup interrogation suggested that BMPR1A mutations were more frequent in early-onset H/L patients receiving FOLFOX compared with other ancestry–treatment groups. Upon formal testing, the case cohort (EOCRC H/L, *n* = 73) and control cohort (EO NHW, *n* = 375) were evaluated for BMPR1A mutation enrichment under the constraint of SMAD4 mutation positivity. Fisher’s exact test revealed no statistically significant difference in BMPR1A mutation prevalence between groups (*p* = 0.836), with an odds ratio of 0.0 (95% CI: 0.033–12.145) (Appendix A). The proportion of in-context BMPR1A mutations was 1.07% in controls versus 0.68% in cases, indicating only a marginal difference in mutation occurrence. These results suggest that while AI-based exploratory scanning flagged BMPR1A as potentially enriched in the EOCRC H/L FOLFOX subgroup, confirmatory statistical testing did not support a significant ancestry-specific association in the context of concurrent SMAD4 alterations.

Third, the AI scan identified multiple clinical and molecular attributes that significantly differed between early-onset NHW and early-onset H/L patients treated with FOLFOX (*p* < 0.05). Statistically significant features included demographic variables (ethnicity group, race), clinical characteristics (primary tumor site, stage at diagnosis, sample type), and disease outcomes (overall survival status, event occurrence). In addition, several key driver mutations—including SMAD4, APC, TCF7L2, TP53, PIK3CA, KRAS, and BRAF—showed differential prevalence between cohorts (Appendix A). Notably, tumor location and highest recorded stage emerged as prominent distinguishing features, aligning with prior evidence that anatomical site and disease stage at presentation can vary by ancestry. These findings underscore the ability of AI-enabled interrogation to rapidly surface clinically relevant subgroup distinctions, guiding targeted downstream statistical analyses.

Finally, the system flagged a possible disparity in *SMAD4* mutation prevalence between early-onset H/L and NHW patients treated with FOLFOX. Analysis revealed that *SMAD4* mutations were present in 2.74% of the H/L case cohort (*n* = 73) compared with 13.87% of the NHW control cohort (*n* = 375). This difference was statistically significant (Chi-square *p* = 0.013), with an odds ratio of 0.175 (95% CI: 0.042–0.735), indicating that early-onset H/L patients treated with FOLFOX were markedly less likely to harbor *SMAD4* mutations than their NHW counterparts (Appendix A). These findings suggest potential ancestry-related differences in the molecular landscape of FOLFOX-treated early-onset CRC, which may have implications for tumor biology and treatment response.

These exploratory outputs informed the selection of subgroup comparisons for formal statistical analyses. The AI system then executed automated filtering and cohort construction based on combined clinical, molecular, and treatment parameters, producing validated mutation frequency tables and survival stratifications. This AI-guided workflow reduced manual data handling, ensured reproducibility, and accelerated the transition from hypothesis generation to confirmatory analysis.

## 3. Discussion

This study represents one of the first artificial intelligence agent-enabled precision medicine analyses of TGF-beta pathway alterations in CRC stratified by age at onset, ancestry, and FOLFOX treatment status, with a particular focus on H/L populations disproportionately affected by EOCRC. Leveraging the AI-HOPE-TGF-Beta platform, we integrated multi-dimensional genomic and clinical data to uncover ancestry- and treatment-specific patterns of pathway disruption and survival impact.

Our findings reveal a highly specific association: EOCRC H/L patients treated with FOLFOX who harbored TGF-beta pathway alterations experienced significantly worse overall survival compared to their non-altered counterparts. This association was absent in all other subgroups, including EOCRC H/L patients not receiving FOLFOX, late-onset H/L patients, and both EOCRC and LOCRC NHW cohorts, regardless of treatment status. These results suggest a possible gene–treatment–ancestry interaction, in which TGF-beta pathway dysregulation may confer treatment resistance or promote aggressive tumor biology specifically in young H/L patients exposed to oxaliplatin-based chemotherapy.

### 3.1. Biological Implications of TGF-Beta Pathway Alterations

The TGF-beta signaling pathway plays a dual role in CRC—functioning as a tumor suppressor in early stages and as a pro-metastatic, pro-immune evasion driver in later stages. Alterations in key pathway members, particularly SMAD4, TGFBR2, SMAD2, and SMAD3, may shift the pathway toward oncogenic activity, promoting epithelial-to-mesenchymal transition (EMT), immune suppression, and metastatic spread. The predominance of SMAD4 alterations across all ancestry and age groups in our dataset aligns with prior evidence linking SMAD4 loss to advanced disease, chemoresistance, and poor prognosis. Notably, our study identifies BMPR1A mutations as disproportionately enriched in EO H/L patients treated with FOLFOX—a finding not previously reported—which may indicate additional ancestry-linked biology within the TGF-beta superfamily.

### 3.2. Ancestry-Specific Genomic Patterns and Treatment Context

While overall TGF-beta pathway alteration frequencies were similar between H/L and NHW patients in EOCRC, our analyses revealed notable exceptions. In LOCRC, H/L patients treated with FOLFOX exhibited a higher prevalence of TGF-beta alterations than their NHW counterparts, whereas NHW late-onset patients receiving FOLFOX had a significantly lower prevalence than those not treated. This suggests that both tumor genomic architecture and treatment selection pressures may differ across ancestry groups. These differences could reflect underlying germline variants influencing somatic evolution, environmental exposures, or healthcare access patterns that shape tumor biology and treatment outcomes.

### 3.3. Implications for FOLFOX Response in EO H/L Patients

The most clinically actionable result from this study is the survival disadvantage observed in FOLFOX-treated EO H/L patients with TGF-beta pathway alterations. Preclinical evidence suggests that TGF-beta activation may drive chemoresistance through multiple mechanisms, including modulation of DNA damage repair, induction of cancer stem cell phenotypes, and reshaping of the tumor microenvironment to suppress anti-tumor immunity. The specificity of the survival effect to EO H/L patients could indicate that host–tumor interactions, potentially shaped by genomic ancestry, play a critical role in modulating chemotherapy benefit. If validated, TGF-beta pathway status could serve as a biomarker to guide therapy selection in this high-risk group, potentially favoring non–oxaliplatin-based regimens or combination strategies incorporating TGF-beta inhibitors currently in development.

Our findings have potential implications for clinical decision-making, particularly in the management of early-onset Hispanic/Latino (H/L) colorectal cancer patients treated with FOLFOX. The observation that TGF-β pathway alterations, especially in SMAD4 and BMPR1A, are associated with significantly poorer survival in this subgroup suggests that pathway status may serve as an ancestry- and treatment-specific prognostic biomarker. While these results are exploratory and require validation in larger and independent cohorts, they highlight the importance of considering molecular features alongside ancestry and treatment context when evaluating prognosis. If confirmed, TGF-β pathway alterations could inform treatment selection by identifying patients who may derive less benefit from oxaliplatin-based chemotherapy and might instead be candidates for alternative regimens or enrollment in clinical trials testing TGF-β–targeted agents. Thus, our study underscores the need to integrate genomic stratification into precision oncology strategies for disproportionately affected populations.

Beyond its prognostic significance, our work also raises the possibility that TGF-β pathway alterations could be leveraged to guide treatment selection and clinical trial design. The observed survival disadvantage in FOLFOX-treated EO H/L patients with pathway alterations suggests that oxaliplatin-based regimens may not provide optimal benefit in this subgroup. If validated, pathway status could therefore serve as a stratification factor in therapeutic decision-making—identifying patients who may benefit from alternative chemotherapy backbones or emerging targeted approaches. In particular, the growing pipeline of TGF-β inhibitors and combination strategies aimed at overcoming immune suppression and chemoresistance offers an opportunity to test whether patients with pathway alterations derive improved outcomes when these agents are integrated. Designing trials that specifically include EO H/L CRC patients and incorporate ancestry-informed stratification will be critical to ensuring both biological and clinical generalizability. Thus, our findings provide a rationale for future translational studies that bridge genomic discovery with precision therapeutic interventions.

Our findings also have important translational implications. Clinically, the observation that TGF-β pathway alterations predict poorer survival in FOLFOX-treated EO H/L patients suggests that pathway status may serve as a biomarker to guide therapeutic decisions in this high-risk group. If validated, these alterations could inform consideration of non-oxaliplatin-based regimens or the inclusion of patients in clinical trials testing TGF-β inhibitors and other targeted agents. Biologically, the results reinforce the dual role of TGF-β signaling in colorectal cancer: functioning as a tumor suppressor in early stages, but shifting toward pro-metastatic, immune-evasive, and chemoresistant roles when disrupted through alterations in key regulators such as SMAD4 and BMPR1A. Together, these insights underscore how integrating ancestry, treatment context, and pathway-specific genomic information can enhance the translational relevance of biomarker discovery and inform precision medicine strategies for disproportionately affected populations.

Finally, by integrating ancestry, treatment context, and pathway-specific alterations, our study provides a framework for linking genomic discoveries to clinical application. The enrichment of SMAD4 and BMPR1A mutations in EO H/L patients treated with FOLFOX suggests a subgroup that may particularly benefit from risk stratification and prioritization in trials of emerging TGF-β inhibitors or alternative regimens. Mechanistically, these alterations likely contribute to shifting TGF-β signaling from tumor-suppressive to pro-oncogenic activity, consistent with its established role in promoting epithelial-to-mesenchymal transition, immune evasion, and chemoresistance. Together, these insights enhance the translational value of our work and underscore the importance of validating these associations in larger, ancestrally diverse patient cohorts.

### 3.4. AI-HOPE-TGF-Beta as an Enabling Technology

The application of AI-HOPE-TGF-Beta was pivotal in uncovering these insights. By automating genomic curation, harmonizing multi-institutional clinical data, and enabling stratified analyses across multiple demographic and treatment dimensions, this platform overcomes a major barrier in precision oncology: the complexity of integrating heterogeneous datasets at scale. Importantly, AI-driven approaches allowed us to rapidly test hypotheses across numerous biologically and clinically relevant strata, a process that would be prohibitively time-consuming with manual workflows.

In addition to survival analyses, AI-enabled exploratory interrogation provided several key pre-statistical insights that shaped downstream hypothesis testing. The AI-HOPE-TGF-Beta platform rapidly identified a significant survival disadvantage among EO H/L CRC patients treated with FOLFOX who harbored TGF-β pathway alterations, a finding subsequently validated by Kaplan–Meier analysis (*p* = 0.029) and consistent with the survival patterns reported in our primary analyses. While AI-driven scans flagged BMPR1A mutations as potentially enriched in this subgroup, confirmatory testing under the constraint of *SMAD4* mutation positivity revealed no statistically significant ancestry-specific association. The platform also highlighted multiple demographic, clinical, and molecular features that differed significantly between EO H/L and EO NHW patients receiving FOLFOX, including disparities in primary tumor site, stage at diagnosis, and prevalence of key driver mutations (*SMAD4*, *APC*, *TCF7L2*, *TP53*, *PIK3CA*, *KRAS*, and *BRAF*), underscoring ancestry-associated heterogeneity in disease presentation. Notably, AI-guided comparison revealed that *SMAD4* mutations were markedly less frequent in EO H/L patients (2.74%) than in EO NHW patients (13.87%) treated with FOLFOX (*p* = 0.013; OR = 0.175, 95% CI: 0.042–0.735), suggesting potential ancestry-related molecular differences with implications for treatment response. These AI-generated insights not only directed the selection of subgroup comparisons for formal statistical testing but also exemplified the platform’s ability to accelerate precision oncology research by automating cohort construction, reducing manual data handling, and ensuring reproducibility.

### 3.5. Limitations and Future Directions

Our study has several limitations. First, despite the large overall sample size, some stratified subgroups—particularly EO H/L patients with pathway alterations—had limited case counts, resulting in wider confidence intervals and reduced statistical power. Second, treatment data were limited to FOLFOX status, without granular details on dosing, duration, or use of targeted agents, which could further refine outcome associations. Third, functional validation of specific mutations was beyond the scope of this analysis, leaving the mechanistic basis for the observed survival effect to future laboratory studies. Finally, our results require validation in independent, ancestrally diverse cohorts to confirm generalizability and assess the utility of TGF-beta pathway status as a predictive biomarker. Future directions include incorporating state-of-the-art molecular characterization approaches such as spatial biology [35] and single-cell sequencing [36] to dissect the tumor microenvironment, resolve cellular heterogeneity, and elucidate spatially organized signaling events that may underlie ancestry- and treatment-specific outcomes.

An important challenge of this study is the relatively small number of early-onset H/L patients with TGF-β pathway alterations, which reduces statistical power and contributes to wider confidence intervals in subgroup analyses. As such, these results should be interpreted as exploratory and hypothesis-generating rather than definitive. Nonetheless, this work leverages one of the only available large-scale datasets integrating clinical and genomic data from H/L colorectal cancer patients, a historically underrepresented population in cancer genomics. Validation in larger, ancestrally diverse cohorts will be essential to confirm these findings and assess their clinical utility.

A limitation of this work is the use of surname-based algorithms to help assign ethnicity, which carries the potential for misclassification bias. While we minimized this concern by using validated surname–ethnicity linkages and harmonized clinical annotations from TCGA, MSK-IMPACT, and AACR Project GENIE, we recognize that such methods are imperfect. Any potential misclassification would likely attenuate rather than exaggerate ancestry-specific differences, suggesting our findings represent conservative estimates. Future studies integrating genetic ancestry inference alongside self-reported ethnicity will be critical to refine classification and further strengthen ancestry-specific precision medicine approaches in disproportionately affected populations.

Our findings also raise important translational considerations for the design of future clinical trials. The consistent association between TGF-β pathway alterations and poor survival in FOLFOX-treated EO H/L patients highlights the potential value of incorporating ancestry, treatment exposure, and molecular context into trial stratification strategies. In particular, prospective studies of TGF-β pathway inhibitors may benefit from prioritizing or enriching for EO H/L patients receiving oxaliplatin-based therapy, where the prognostic impact appears most pronounced. More broadly, these results underscore how ancestry- and treatment-informed biomarker discovery can guide the development of precision therapeutics and ensure that disproportionately affected populations are meaningfully represented in emerging clinical interventions [37,38].

It is important to emphasize that the associations observed between TGF-β pathway alterations and survival in FOLFOX-treated EO H/L patients reflect prognostic correlations rather than deterministic causal mechanisms. While these findings highlight the potential of TGF-β alterations as biomarkers, the biological underpinnings linking these mutations to treatment-specific outcomes remain to be clarified. Accordingly, our results should be viewed as hypothesis-generating, warranting further validation through functional studies, preclinical modeling, and integration of complementary multi-omics datasets to uncover mechanistic drivers.

Our results also underscore an important direction for future research: whether biomarker status should inform treatment adaptation in EO H/L CRC. The observation that TGF-β pathway alterations predict poorer survival in patients receiving FOLFOX suggests that alternative therapeutic approaches may be warranted in this subgroup. Prospective studies that incorporate biomarker-guided trial stratification—such as evaluating intensified chemotherapy regimens, adding targeted agents including TGF-β pathway inhibitors, or testing novel combination strategies—will be essential to determine if modifying therapy based on pathway alterations can improve clinical outcomes.

A key strength of this study is that it leverages one of the few available large-scale databases integrating clinical and genomic data from Hispanic/Latino colorectal cancer patients, a population historically underrepresented in cancer genomics research. This unique resource enabled us to uncover ancestry- and treatment-specific prognostic associations that may otherwise remain invisible in predominantly NHW-focused datasets. At the same time, we acknowledge that the absence of functional validation limits mechanistic interpretation of these findings. Future work should include experimental interrogation of SMAD4, BMPR1A, and other TGF-β pathway alterations in preclinical models, as well as advanced approaches such as single-cell sequencing and spatial biology to dissect tumor–microenvironment interactions. These efforts will be critical to confirm the biological basis of the observed survival disadvantage in FOLFOX-treated EO H/L patients and to translate our findings into clinically actionable strategies.

Future work will be essential to functionally validate the prognostic role of TGF-β pathway alterations identified in this study. Experimental approaches such as CRISPR-based knockout or knock-in of SMAD4 and BMPR1A in colorectal cancer cell lines and patient-derived organoids could directly test their effects on chemoresistance and tumor progression. Complementary use of spatial biology and single-cell transcriptomics will further clarify how these alterations shape the tumor microenvironment and treatment response, particularly in early-onset H/L patients receiving FOLFOX. These mechanistic studies will be critical to move from associative observations toward causal understanding, thereby strengthening the translational potential of TGF-β pathway alterations as prognostic biomarkers and therapeutic targets.

## 4. Materials and Methods

### 4.1. Data Sources and Cohort Assembly

We conducted a retrospective analysis using de-identified clinical and genomic data from three publicly accessible CRC datasets available through the cBioPortal for Cancer Genomics platform: Colorectal Adenocarcinoma (TCGA, PanCancer Atlas), MSK-CHORD (MSK, Nature 2024), and GENIE BPC CRC. Datasets were selected for their inclusion of both somatic variant profiles and detailed therapeutic annotations, enabling accurate identification of chemotherapy regimens. Eligible cases included histologically confirmed colorectal, colon, or rectal adenocarcinoma with available primary tumor sequencing data. When multiple tumor samples were present for a single individual, only one sample was retained to avoid duplication bias.

FOLFOX treatment status was defined based on the presence of medication combinations consistent with oxaliplatin, fluorouracil, and leucovorin therapy, as annotated in the cBioPortal datasets. Detailed information regarding dosing schedules, treatment duration, or concurrent targeted/biologic therapies (e.g., cetuximab, bevacizumab) was not uniformly available and therefore not included in this analysis. Patients were thus classified dichotomously as “FOLFOX-treated” or “non-FOLFOX” for the purposes of survival and genomic stratification.

### 4.2. Identification of Disproportionately Affected Populations

Ethnicity classification prioritized self-reported annotations within the original datasets. Individuals were assigned to the H/L group if labeled as “Hispanic or Latino,” “Spanish, NOS,” “Hispanic, NOS,” or “Latino, NOS.” When ethnicity was not explicitly stated, surname-based classification was applied using validated algorithms for identifying Hispanic origin. The comparator group consisted of Non-Hispanic White (NHW) patients meeting the same inclusion criteria. Age at diagnosis was extracted from clinical metadata, with EOCRC defined as diagnosis before age 50 and LOCRC as diagnosis at age 50 or older.

### 4.3. Treatment Classification

Patients were categorized as “FOLFOX-treated” if their treatment records documented concurrent or sequential administration of leucovorin, fluorouracil (5-FU), and oxaliplatin, consistent with standard first-line protocols for metastatic microsatellite-stable CRC. Treatment start and stop dates were cross-referenced to confirm overlapping timelines for the three drugs. Individuals without recorded use of all three components were assigned to the non-FOLFOX group.

### 4.4. Definition of TGF-Beta Pathway Alterations

A curated list of TGF-beta signaling genes was compiled from peer-reviewed literature and pathway databases [24,25,26,27,28,29]. Genes included SMAD family members, TGFBR1/2, BMP ligands, and other canonical signaling components implicated in CRC development and progression. Somatic alterations were extracted from cBioPortal mutation data and filtered to retain only non-synonymous variants (missense, nonsense, frameshift insertions/deletions, splice site, and start codon mutations). Pathway alteration status was defined as the presence of at least one qualifying mutation in any of the pathway genes.

### 4.5. Statistical Analysis

Comparisons of mutation frequencies between groups were performed using Fisher’s exact test or chi-square test, as appropriate. Continuous variables were evaluated with the Mann–Whitney U test. Overall survival (OS) was assessed using the Kaplan–Meier method, and differences between survival curves were evaluated using the log-rank test. Hazard ratios (HRs) and 95% confidence intervals (CIs) were estimated via univariate and multivariate Cox proportional hazards regression models. All statistical analyses were conducted in R (v4.3.2), with *p*-values < 0.05 considered statistically significant. Multivariate Cox proportional hazards regression models were constructed to adjust for ancestry, treatment status, comorbidities (when available), microsatellite instability (MSI) status, original tumor site, and tumor stage, in addition to TGF-β pathway alteration status.

### 4.6. Artificial Intelligence-Enabled Data Integration, Post-Analysis Scanning, and Query Execution

To enhance analytical efficiency, ensure reproducibility, and strategically guide downstream statistical testing, we first deployed the AI-HOPE-TGF-Beta conversational AI agent [34]—a specialized module within the AI-HOPE precision medicine framework [33]—to conduct a comprehensive post-analysis scan of the selected CRC datasets. This process leveraged targeted, natural language-driven queries to rapidly interrogate the data and identify clinically relevant patterns for formal evaluation. Representative queries included: Among H/L CRC patients treated with FOLFOX, does TGF-β pathway alteration status associate with overall survival (OS)? Is BMPR1A mutation frequency different among early-onset H/L patients treated with FOLFOX? Identify all clinical features associated with NHW patients with TGF-β pathway alterations versus those without. Is there a difference in SMAD4 mutation prevalence between EOCRC H/L patients treated with FOLFOX and EOCRC NHW CRC patients treated with FOLFOX?

The AI-HOPE [33] and AI-HOPE-TGF-Beta [34] platforms integrate structured clinical, genomic, and treatment data, enabling automated filtering, subgroup stratification, and statistical comparison based on these high-priority questions. Following this preliminary scan, the AI systems were used to (1) formally identify patients meeting combined clinical (EOCRC, treatment group, ancestry) and molecular (TGF-beta pathway status) criteria; (2) generate subgroup-specific mutation frequency tables; and (3) perform outcome-based stratifications for survival analyses. The conversational interface supported iterative refinement of query parameters, ensuring alignment with study objectives, reducing manual data handling errors, and accelerating the transition from exploratory interrogation to validated statistical testing.

All outputs generated through AI-HOPE and AI-HOPE-TGFβ were cross-validated against results obtained using standard bioinformatics pipelines (Fisher’s exact test, chi-square test, and Kaplan–Meier survival analysis) to ensure reproducibility and concordance with established statistical methods.

## 5. Conclusions

This AI-enabled, precision oncology analysis identifies TGF-beta pathway alterations as a potential biomarker of poor survival specifically in EO H/L CRC patients receiving FOLFOX chemotherapy. These findings highlight the importance of integrating ancestry, age, and treatment context into genomic outcome studies and suggest a path toward more personalized treatment strategies for disproportionately affected populations. By revealing ancestry-specific vulnerabilities and treatment interactions, this work underscores the transformative potential of artificial intelligence in advancing health equity in cancer genomics.

## Figures and Tables

**Figure 1 ijms-26-09067-f001:**
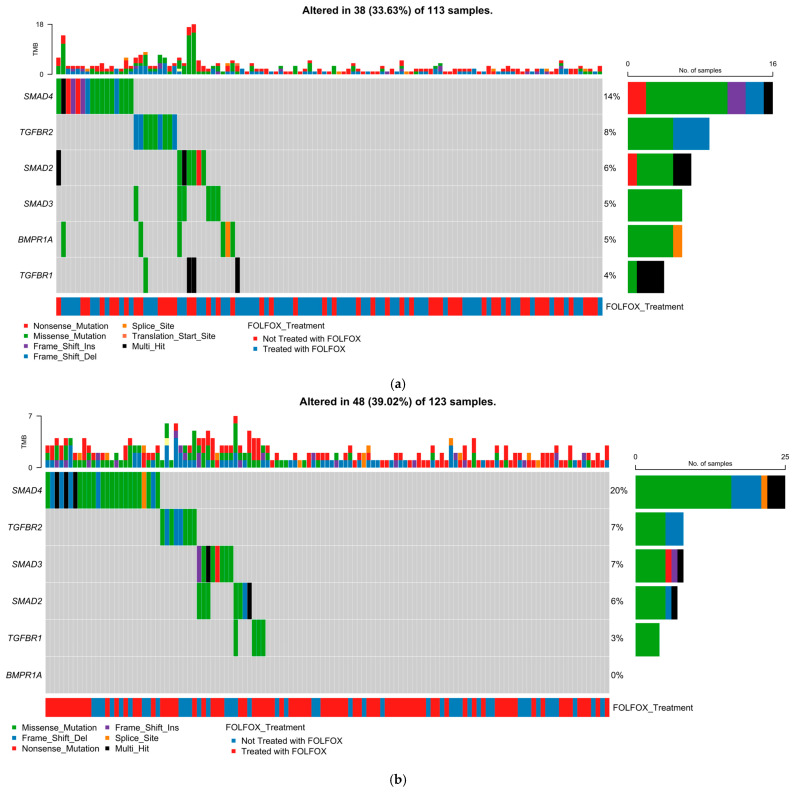
Somatic mutation landscape of TGF-beta pathway genes in colorectal cancer (CRC) stratified by age of onset and ancestry. Oncoplots showing gene-level mutation profiles of key TGF-beta pathway components (SMAD4, TGFBR2, SMAD2, SMAD3, BMPR1A, and TGFBR1) in colorectal cancer, stratified by age of onset (early vs. late) and ancestry (Hispanic/Latino vs. Non-Hispanic White). Panels display mutation types, tumor mutational burden (TMB), and FOLFOX treatment status across: (**a**) 113 early-onset Hispanic/Latino (H/L) patients, (**b**) 123 late-onset H/L patients, (**c**) 607 early-onset Non-Hispanic White (NHW) patients, and (**d**) 1409 late-onset NHW patients. Across all subgroups, SMAD4 is the most frequently mutated gene, with missense variants predominating, followed by recurrent alterations in TGFBR2, SMAD2, and SMAD3. Lower-frequency events are observed in BMPR1A and TGFBR1. The data highlight the widespread disruption of TGF-beta signaling in CRC and suggest potential age- and ancestry-associated variation in the somatic mutation landscape, as well as representation across both FOLFOX-treated and untreated patients.

**Figure 2 ijms-26-09067-f002:**
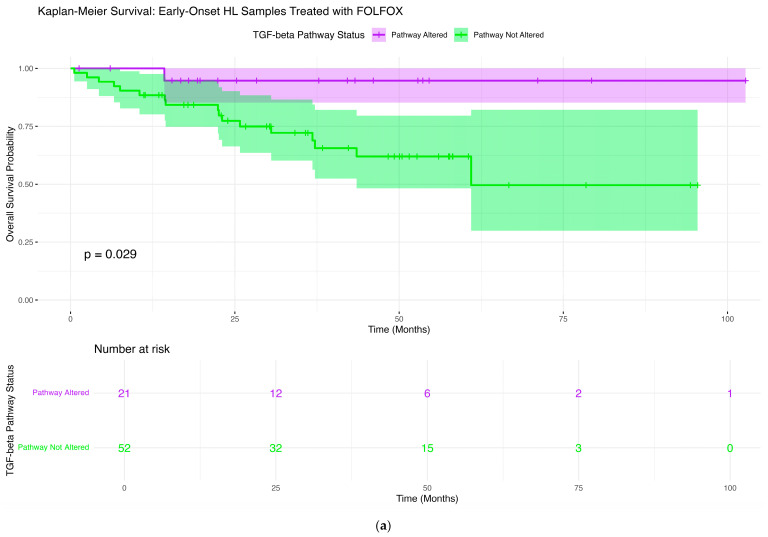
Kaplan–Meier survival analysis of TGF-beta pathway alterations across colorectal cancer (CRC) subgroups defined by age, ancestry, and FOLFOX treatment status. Overall survival curves are shown for (**a**) Early-Onset Hispanic/Latino (H/L) treated with FOLFOX, (**b**) Early-Onset H/L not treated with FOLFOX, (**c**) Late-Onset H/L treated with FOLFOX, (**d**) Late-Onset H/L not treated with FOLFOX, (**e**) Early-Onset Non-Hispanic White (NHW) treated with FOLFOX, and (**f**) Early-Onset NHW not treated with FOLFOX. Each plot compares patients with TGF-beta pathway alterations to those without, illustrating subgroup-specific differences in survival outcomes. Shaded areas represent 95% confidence intervals, and number-at-risk tables indicate patient counts at key follow-up intervals.

**Table 1 ijms-26-09067-t001:** Clinical and demographic characteristics of Hispanic/Latino (H/L) and Non-Hispanic White (NHW) colorectal cancer (CRC) patients, stratified by age at diagnosis, FOLFOX treatment status, tumor features, and ethnicity.

Clinical Feature	H/L Cohort n (%)	NHW Cohort n (%)
Patients with and without FOLFOX Treatment		
Treated with FOLFOX	164 (61.6%)	1294 (57.6%)
Not Treated with FOLFOX	102 (38.3%)	955 (42.4%)
Cancer Type		
Colorectal Adenocarcinoma	266 (100%)	2249 (100%)
Sex		
Male	158 (59.4%)	1267 (56.3%)
Female	108 (40.6%)	982 (43.7%)
Stage at Diagnosis		
Stage 1–3	156 (58.6%)	1236 (55.0%)
Stage 4	108 (40.6%)	1005 (44.7%)
NA	2 (0.8%)	8 (0.4%)
MSI Type		
Stable	200 (75.2%)	1940 (86.3%)
Instable	21 (7.9%)	209 (9.3%)
Indeterminate	10 (3.8%)	57 (2.5%)
NA	35 (13.2%)	43 (1.9%)
Ethnicity		
Spanish, Hispanic, Latino NOS	236 (88.8%)	0 (0.0%)
Mexican (includes Chicano)	30 (11.3%)	0 (0.0%)
Non-Spanish; Non-Hispanic	0 (0.0%)	2249 (100.0%)

**Table 2 ijms-26-09067-t002:** Comparative clinical and genomic profiles of early-onset and late-onset colorectal cancer (CRC) patient cohorts. This table outlines clinical and molecular distinctions, including TGF-Beta pathway alterations and mutation burden, across key subgroups: (**a**) Early-Onset CRC (EOCRC) versus Late-Onset CRC (LOCRC) among Hispanic/Latino (H/L) patients; (**b**) EOCRC versus LOCRC among Non-Hispanic White (NHW) patients; (**c**) EOCRC comparisons between H/L and NHW cohorts; and (**d**) EOCRC versus LOCRC comparisons between H/L treated and not treated with FOLFOX. Comparisons include median age at diagnosis, total mutation counts, and the prevalence of selected TGF-Beta pathway gene alterations, stratified by both ethnicity and age category.

(a)						
Clinical Feature	Early-Onset Hispanic/Latino Treated with FOLFOX n (%)	Early-Onset Hispanic/Latino Not Treated with FOLFOX n (%)	*p*-Value	Late-Onset Hispanic/Latino Treated with FOLFOX n (%)	Late-Onset Hispanic/Latino Not Treated with FOLFOX n (%)	*p*-Value
Median Diagnosis Age (IQR)	42 (36–47)	40 (34–43)	0.05411	59 (54–66)	62 (56–70)	0.04865
Median Mutation Count	7 (5–8)	7 (5–20)	0.09735	8 (6–9) [NA = 1]	7 (5.25–9)	0.6507
Median TMB (IQR)	6.3 (4.5–7.8) [NA = 15]	5.5 (3.4–8.3) [NA = 2]	0.1719	6.1 (4.9–7.8) [NA = 10]	6.9 (5.6–9.0) [NA = 2]	0.04389
Median FGA	0.18 (0.03–0.27) [NA = 6]	0.19 (0.03–0.29)	0.7661	0.15 (0.06–0.25) [NA = 7]	0.21 (0.04–0.3) [NA = 2]	0.5464
BMPR1A Mutation						
Present	4 (5.5%)	2 (3.8%)	1	0 (0.0%)	0 (0.0%)	1
Absent	69 (94.5%)	50 (96.2%)		91 (100.0%)	50 (100.0%)	
**(b)**						
**Clinical Feature**	**Early-Onset NHW** **Treated with FOLFOX** **n (%)**	**Early-Onset NHW** **Not Treated with FOLFOX** **n (%)**	***p*-Value**	**Late-Onset NHW** **Treated with FOLFOX** **n (%)**	**Late-Onset NHW** **Not Treated with FOLFOX** **n (%)**	***p*-Value**
Median Diagnosis Age (IQR)	43 (37–48)	44 (38–47)	0.5646	63 (57–69)	66 (57–74)	4.146 × 10^−7^
Median Mutation Count	6 (5–8) [NA = 4]	7 (5–9) [NA = 2]	0.1258	7 (5–9) [NA = 10]	8 (6–12) [NA = 3]	1.22 × 10^−5^
Median TMB (IQR)	5.7 (4.1–6.9)	5.7 (4.1–7.8)	0.4214	6.1 (4.3–8.2)	6.6 (4.9–10.4)	0.0002854
Median FGA	0.14 (0.04–0.24) [NA = 4]	0.15 (0.04–0.23) [NA = 2]	0.5589	0.16 (0.06–0.28) [NA = 6]	0.15 (0.05–0.27) [NA = 5]	0.1929
BMPR1A Mutation						
Present	4 (1.1%)	10 (3.3%)	0.05575	15 (1.6%)	15 (2.3%)	0.4458
Absent	371 (98.9%)	292 (96.7%)		904 (98.4%)	638 (97.7%)	
SMAD2 Mutation						
Present	19 (5.1%)	16 (5.3%)	1	35 (3.8%)	43 (6.6%)	0.0173
Absent	356 (94.9%)	286 (94.7%)		884 (96.2%)	610 (93.4%)	
SMAD3 Mutation						
Present	14 (3.7%)	12 (4.0%)	1	29 (3.2%)	34 (5.2%)	0.05578
Absent	361 (96.3%)	290 (96.0%)		890 (96.8%)	619 (94.8%)	
TGFBR2 Mutation						
Present	11 (2.9%)	16 (5.3%)	0.1721	38 (4.1%)	46 (7.0%)	0.01578
Absent	364 (97.1%)	286 (94.7%)		881 (95.9%)	607 (93.0%)	
**(c)**						
**Clinical Feature**	**Early-Onset Hispanic/Latino** **Treated with FOLFOX** **n (%)**	**Early-Onset NHW** **Treated with FOLFOX** **n (%)**	***p*-Value**	**Early-Onset Hispanic/Latino** **Not Treated with FOLFOX** **n (%)**	**Early-Onset NHW** **Not Treated with FOLFOX** **n (%)**	***p*-Value**
Median Diagnosis Age (IQR)	42 (36–47)	43 (37–48)	0.08467	40 (34–43)	44 (38–47)	0.0006016
Median Mutation Count	7 (5–8)	6 (5–8) [NA = 4]	0.942	7 (5–20)	7 (5–9) [NA = 2]	0.2601
Median TMB (IQR)	6.3 (4.5–7.8) [NA = 15]	5.7 (4.1–6.9)	0.05806	5.5 (3.4–8.3) [NA = 2]	5.7 (4.1–7.8)	0.5732
Median FGA	0.18 (0.03–0.27) [NA = 6]	0.14 (0.04–0.24) [NA = 4]	0.5556	0.19 (0.03–0.29)	0.15 (0.04–0.23) [NA = 2]	0.3612
BMPR1A Mutation						
Present	4 (5.5%)	4 (1.1%)	0.02715	2 (3.8%)	10 (3.3%)	0.6916
Absent	69 (94.5%)	371 (98.9%)		50 (96.2%)	292 (96.7%)	
**(d)**						
**Clinical Feature**	**Early-Onset Hispanic/Latino** **Treated with FOLFOX** **n (%)**	**Late-Onset Hispanic/Latino** **Treated with FOLFOX** **n (%)**	***p*-Value**	**Early-Onset Hispanic/Latino** **Not Treated with FOLFOX** **n (%)**	**Late-Onset Hispanic/Latino** **Not Treated with FOLFOX** **n (%)**	***p*-Value**
Median Diagnosis Age (IQR)	42 (36–47)	59 (54–66)	2.20 × 10^−16^	40 (34–43)	62 (56–70)	2.20 × 10^−16^
Median Mutation Count	7 (5–8)	8 (6–9) [NA = 1]	0.01715	7 (5–20)	7 (5.25–9)	0.7701
Median TMB (IQR)	6.3 (4.5–7.8) [NA = 15]	6.1 (4.9–7.8) [NA = 10]	0.6472	5.5 (3.4–8.3) [NA = 2]	6.9 (5.6–9.0) [NA = 2]	0.03283
Median FGA	0.18 (0.03–0.27) [NA = 6]	0.15 (0.06–0.25) [NA = 7]	0.9746	0.19 (0.03–0.29)	0.21 (0.04–0.3) [NA = 2]	0.7905
BMPR1A Mutation						
Present	4 (5.5%)	0 (0.0%)	0.03747	2 (3.8%)	0 (0.0%)	0.4952
Absent	69 (94.5%)	91 (100.0%)		50 (96.2%)	50 (100.0%)	

**Table 3 ijms-26-09067-t003:** Frequency of TGF-beta pathway alterations in colorectal cancer (CRC) patients stratified by age of onset, ancestry, and FOLFOX treatment status. This table summarizes the mutation frequencies of key genes involved in the TGF-beta signaling pathway among CRC patients. Analyses are stratified as follows: (**a**) early-onset (EOCRC) vs. late-onset (LOCRC) and FOLFOX treatment status within the Hispanic/Latino (H/L) cohort; (**b**) FOLFOX-treated vs. untreated patients within EOCRC and LOCRC subgroups of Non-Hispanic White (NHW) patients; (**c**) EOCRC H/L vs. NHW patients by FOLFOX treatment status; and (**d**) LOCRC H/L vs. NHW patients by FOLFOX treatment status. Genes analyzed include BMPR1A, SMAD2, SMAD3, SMAD4, TGFBR1, and TGFBR2. Statistically significant differences (*p* < 0.05, Chi-square or Fisher’s exact test) are indicated with asterisks. This stratified analysis highlights potential interactions between age, ancestry, chemotherapy exposure, and TGF-beta pathway dysregulation.

(a)						
Pathway Alterations	Early-Onset Hispanic/Latino Treated with FOLFOX n (%)	Early-Onset Hispanic/Latino Not Treated with FOLFOX n (%)	*p*-Value	Late-Onset Hispanic/Latino Treated with FOLFOX n (%)	Late-Onset Hispanic/Latino Not Treated with FOLFOX n (%)	*p*-Value
TGF-beta Alterations Present	21 (28.8%)	19 (36.5%)	0.4693	30 (33.0%)	19 (38.0%)	0.6777
TGF-beta Alterations Absent	52 (71.2%)	33 (63.5%)		61 (67.0%)	31 (62.0%)	
**(b)**						
**Pathway Alterations**	**Early-Onset NHW** **Treated with FOLFOX** **n (%)**	**Early-Onset NHW** **Not Treated with FOLFOX** **n (%)**	***p*-Value**	**Late-Onset NHW** **Treated with FOLFOX** **n (%)**	**Late-Onset NHW** **Not Treated with FOLFOX** **n (%)**	***p*-Value**
TGF-beta Alterations Present	92 (24.5%)	79 (26.2%)	0.6929	215 (23.4%)	205 (31.4%)	0.0005127
TGF-beta Alterations Absent	283 (75.5%)	223 (73.8%)		704 (76.6%)	448 (68.6%)	
**(c)**						
**Pathway Alterations**	**Early-Onset Hispanic/Latino** **Treated with FOLFOX** **n (%)**	**Early-Onset NHW** **Treated with FOLFOX** **n (%)**	***p*-Value**	**Early-Onset Hispanic/Latino** **Not Treated with FOLFOX** **n (%)**	**Early-Onset NHW** **Not Treated with FOLFOX** **n (%)**	***p*-Value**
TGF-beta Alterations Present	21 (28.8%)	92 (24.5%)	0.5387	19 (36.5%)	79 (26.2%)	0.1684
TGF-beta Alterations Absent	52 (71.2%)	283 (75.5%)		33 (63.5%)	223 (73.8%)	
**(d)**						
**Pathway Alterations**	**Late-Onset Hispanic/Latino** **Treated with FOLFOX** **n (%)**	**Late-Onset NHW** **Treated with FOLFOX** **n (%)**	***p*-Value**	**Late-Onset Hispanic/Latino** **Not Treated with FOLFOX** **n (%)**	**Late-Onset NHW** **Not Treated with FOLFOX** **n (%)**	***p*-Value**
TGF-beta Alterations Present	30 (33.0%)	215 (23.4%)	0.05693	19 (38.0%)	205 (31.4%)	0.4186
TGF-beta Alterations Absent	61 (67.0%)	704 (76.6%)		31 (62.0%)	448 (68.6%)	

## Data Availability

All data used in the present study is publicly available at https://www.cbioportal.org/ (accessed on 4 July 2025) and https://genie.cbioportal.org (accessed on 4 July 2025). Additional data can be provided upon reasonable request to the authors.

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
