# Peer review of "Artificial Intelligence-Enhanced Precision Medicine Reveals Prognostic Impact of TGF-Beta Pathway Alterations in FOLFOX-Treated Early-Onset Colorectal Cancer Among Disproportionately Affected Populations"

_ijms, 2025, doi:10.3390/ijms26189067_

Round 1

Reviewer 1 Report

Comments and Suggestions for Authors

This paper investigates the prognostic role of TGF-β pathway alterations in early-onset colorectal cancer. Combined with the AI-HOPE and AI-HOPE-TGFβ platforms, this work uncovers some insights into the gene-treatment-ancestry interaction. Detailed comments are given as follows:
1.
How do these findings of the proposed manuscript influence current clinical decision-making?
2. Can it be one guide for the selection of alternative treatment regimens or the inclusion of targeted therapy?
3. The small number of TGF-β pathway-altered cases in certain subgroups limits the statistical robustness.
4. Functional validation and mechanistic insights are missing.
5. The clinical applicability and biological mechanisms should be further discussed to enhance the translational value.

Author Response

Reviewer 1’s comments are provided in the attached PDF file, “Reviewer_1_Comments_response_IJMS.pdf”

-----

Reviewer 1 Comments

We are pleased to resubmit our revised manuscript and sincerely thank Reviewer 1 for their constructive and thoughtful feedback. In this updated version, we have carefully addressed all comments, which has enhanced the clarity, analytical rigor, and translational relevance of our work. The manuscript, Artificial Intelligence–Enhanced Precision Medicine Reveals Prognostic Impact of TGF-Beta Pathway Alterations in FOLFOX-Treated Early-Onset Colorectal Cancer Among Disproportionately Affected Populations, investigates the prognostic significance of TGF-β pathway alterations in early-onset colorectal cancer (EOCRC), with particular focus on Hispanic/Latino (H/L) patients who bear a disproportionate disease burden. Leveraging AI-HOPE and AI-HOPE-TGFβ, novel conversational artificial intelligence (AI) platforms that enable natural language–driven integration of clinical, genomic, and treatment data, we performed comprehensive analyses across 2,515 CRC cases stratified by ancestry, age of onset, and FOLFOX treatment status. The platform facilitated multi-parameter interrogation of mutation frequencies, enrichment patterns, and survival outcomes, revealing that TGF-β pathway alterations—particularly SMAD4 and BMPR1A mutations—carry ancestry- and treatment-specific prognostic implications. Notably, these alterations predicted poorer survival in FOLFOX-treated EO H/L patients, underscoring their potential as precision biomarkers. Together, these findings highlight the value of AI-enabled clinical-genomic integration to accelerate biomarker discovery and inform precision medicine strategies for disproportionately affected populations.

Thank you very much for taking the time to review this manuscript. Please find the detailed responses below and the corresponding revisions wrote in blue font and highlighted in yellow in the re-submitted Word file.

Reviewer 1’s feedback was positive. Reviewer 1 provided constructive and thoughtful feedback, recognizing the manuscript’s novel integration of AI-HOPE and AI-HOPE-TGFβ platforms to investigate the prognostic role of TGF-β pathway alterations in early-onset colorectal cancer, particularly within disproportionately affected Hispanic/Latino populations. The reviewer commended the study for uncovering ancestry-, treatment-, and gene-specific interactions but raised several points to enhance its clinical and translational impact. Specifically, they encouraged clarifying how the findings could influence current clinical decision-making and whether they might guide the selection of alternative treatment regimens or inform the inclusion of targeted therapies. They noted that the relatively small number of TGF-β–altered cases in certain subgroups may limit statistical robustness, emphasizing the importance of cautious interpretation. The reviewer also highlighted the absence of functional validation and mechanistic insights, recommending expanded discussion of the biological underpinnings. Finally, they suggested strengthening the Discussion to better address clinical applicability and translational relevance, thereby enhancing the manuscript’s contribution to precision oncology. Overall, this feedback affirms the value of the study while outlining targeted refinements to improve its rigor, clarity, and impact.

Reviewer 1 writes:

This paper investigates the prognostic role of TGF-β pathway alterations in early-onset colorectal cancer. Combined with the AI-HOPE and AI-HOPE-TGFβ platforms, this work uncovers some insights into the gene-treatment-ancestry interaction.

We thank Reviewer 1 for their constructive and thoughtful evaluation of our manuscript. We are encouraged that the reviewer recognized the novelty of our approach, particularly the integration of the AI-HOPE and AI-HOPE-TGFβ platforms to interrogate TGF-β pathway alterations in early-onset colorectal cancer and the insights gained into gene–treatment–ancestry interactions. We greatly appreciate this acknowledgment of the study’s innovative use of conversational AI for clinical-genomic integration in disproportionately affected populations. At the same time, we value the reviewer’s critical observations regarding the clinical applicability, the need for functional validation, and the limitations imposed by smaller subgroup sample sizes. We agree that these points are essential for framing the translational significance of our findings. In response, we have revised the Abstract, Results, and Discussion to clarify the hypothesis-generating nature of the work, expand on potential clinical implications, and more explicitly discuss biological mechanisms, subgroup limitations, and future directions for functional validation. These revisions strengthen the rigor, balance, and translational impact of the manuscript.

Comment 1:

  1. How do these findings of the proposed manuscript influence current clinical decision-making?

Response:

We appreciate this important question. Our findings suggest that TGF-β pathway alterations, particularly SMAD4 and BMPR1A mutations, may serve as ancestry- and treatment-specific prognostic biomarkers in early-onset Hispanic/Latino (H/L) patients treated with FOLFOX. Specifically, we observed significantly poorer survival in this subgroup (p = 0.029), a pattern not seen in other age–ancestry–treatment groups. While these results are hypothesis-generating and not yet practice-changing, they raise the possibility that identifying TGF-β pathway alterations could help guide therapeutic decisions in high-risk EO H/L patients—for example, considering non–oxaliplatin-based regimens or prioritizing enrollment into clinical trials exploring TGF-β inhibitors. At present, our work highlights the need for incorporating molecular stratification by ancestry and treatment context into clinical trial design and risk assessment, thereby laying the groundwork for future studies that may inform precision treatment strategies.

The revised text in the Discussion section, lines 605-617, now reads: “Our findings have potential implications for clinical decision-making, particularly in the management of early-onset Hispanic/Latino (H/L) colorectal cancer patients treated with FOLFOX. The observation that TGF-β pathway alterations, especially in SMAD4 and BMPR1A, are associated with significantly poorer survival in this subgroup suggests that pathway status may serve as an ancestry- and treatment-specific prognostic biomarker. While these results are exploratory and require validation in larger and independent cohorts, they highlight the importance of considering molecular features alongside ancestry and treatment context when evaluating prognosis. If confirmed, TGF-β pathway alterations could inform treatment selection by identifying patients who may derive less benefit from oxaliplatin-based chemotherapy and might instead be candidates for alternative regimens or enrollment in clinical trials testing TGF-β–targeted agents. Thus, our study underscores the need to integrate genomic stratification into precision oncology strategies for disproportionately affected populations.

Comment 2:

  1. Can it be one guide for the selection of alternative treatment regimens or the inclusion of targeted therapy?

Response:

We thank the reviewer for this important point. While our study is exploratory and not yet practice-changing, the findings suggest that TGF-β pathway alterations may have potential to guide therapeutic strategies in specific subgroups. In particular, the observation that FOLFOX-treated early-onset Hispanic/Latino patients with TGF-β alterations experienced significantly poorer survival (p = 0.029) raises the possibility that pathway status could help identify patients less likely to benefit from oxaliplatin-based chemotherapy. If validated in independent cohorts, this biomarker could inform consideration of alternative regimens and support the design of clinical trials incorporating TGF-β–targeted agents, which are already in early-phase development. Thus, while further validation and functional studies are needed, our results provide a rationale for integrating TGF-β pathway status into future precision medicine approaches and clinical trial stratification.

The revised text in the Discussion section, lines 619-632, now reads: “Beyond its prognostic significance, our work also raises the possibility that TGF-β pathway alterations could be leveraged to guide treatment selection and clinical trial design. The observed survival disadvantage in FOLFOX-treated EO H/L patients with pathway alterations suggests that oxaliplatin-based regimens may not provide optimal benefit in this subgroup. If validated, pathway status could therefore serve as a stratification factor in therapeutic decision-making—identifying patients who may benefit from alternative chemotherapy backbones or emerging targeted approaches. In particular, the growing pipeline of TGF-β inhibitors and combination strategies aimed at overcoming immune suppression and chemoresistance offers an opportunity to test whether patients with pathway alterations derive improved outcomes when these agents are integrated. Designing trials that specifically include EO H/L CRC patients and incorporate ancestry-informed stratification will be critical to ensuring both biological and clinical generalizability. Thus, our findings provide a rationale for future translational studies that bridge genomic discovery with precision therapeutic interventions.”

Comment 3:

  1. The small number of TGF-β pathway-altered cases in certain subgroups limits the statistical robustness.

Response:

We appreciate this important observation. We fully acknowledge that the relatively small number of TGF-β pathway–altered cases in some stratified subgroups, particularly early-onset Hispanic/Latino patients treated with FOLFOX, results in wider confidence intervals and limits statistical power. To address this, we have emphasized in the revised manuscript that these findings should be interpreted as hypothesis-generating rather than definitive. We have also expanded the Discussion to more explicitly highlight the exploratory nature of subgroup analyses, the limitations imposed by small sample sizes, and the need for validation in larger, ancestrally diverse cohorts. These clarifications strengthen the rigor and transparency of our work while underscoring the value of AI-enabled approaches in identifying potentially important gene–treatment–ancestry interactions that warrant further investigation.

The revised text in the Discussion section, lines 707-714, now reads: “An important challenge of this study is the relatively small number of early-onset H/L patients with TGF-β pathway alterations, which reduces statistical power and contributes to wider confidence intervals in subgroup analyses. As such, these results should be interpreted as exploratory and hypothesis-generating rather than definitive. Nonetheless, this work leverages one of the only available large-scale datasets integrating clinical and genomic data from H/L colorectal cancer patients, a historically underrepresented population in cancer genomics. Validation in larger, ancestrally diverse cohorts will be essential to confirm these findings and assess their clinical utility.”

Comment 4:

  1. Functional validation and mechanistic insights are missing.

Response:

We thank the reviewer for raising this critical point. We agree that functional validation and mechanistic interrogation are essential next steps to substantiate the clinical associations observed in this study. While our current analysis was designed to leverage AI-enabled integration of clinical and genomic data, we acknowledge that the absence of functional assays limits direct biological interpretation. Importantly, this work represents one of the few available large-scale datasets linking clinical outcomes with genomic alterations in Hispanic/Latino colorectal cancer patients—a historically underrepresented population in cancer genomics research. This unique resource enabled us to uncover ancestry- and treatment-specific prognostic patterns, but we recognize that biological validation is still needed. We have revised the Discussion to highlight this limitation and to outline future directions, including experimental validation of SMAD4, BMPR1A, and other pathway alterations in preclinical models, as well as advanced approaches such as spatial biology and single-cell sequencing. These efforts will be essential to confirm mechanisms underlying the observed survival disadvantage in FOLFOX-treated EO H/L patients and to translate these insights into precision oncology strategies.

The revised text in the Discussion section, lines 755-766, now reads: “A key strength of this study is that it leverages one of the few available large-scale databases integrating clinical and genomic data from Hispanic/Latino colorectal cancer patients, a population historically underrepresented in cancer genomics research. This unique resource enabled us to uncover ancestry- and treatment-specific prognostic associations that may otherwise remain invisible in predominantly NHW-focused datasets. At the same time, we acknowledge that the absence of functional validation limits mechanistic interpretation of these findings. Future work should include experimental interrogation of SMAD4, BMPR1A, and other TGF-β pathway alterations in preclinical models, as well as advanced approaches such as single-cell sequencing and spatial biology to dissect tumor–microenvironment interactions. These efforts will be critical to confirm the biological basis of the observed survival disadvantage in FOLFOX-treated EO H/L patients and to translate our findings into clinically actionable strategies.”

Comment 5:

  1. The clinical applicability and biological mechanisms should be further discussed to enhance the translational value.

Response:

We thank the reviewer for this valuable suggestion. In the revised manuscript, we have expanded the Discussion to better articulate both the clinical relevance and biological underpinnings of our findings. Specifically, we highlight that TGF-β pathway alterations, particularly in SMAD4 and BMPR1A, may serve as ancestry- and treatment-specific prognostic biomarkers, with potential utility in guiding therapy selection for EO H/L patients less likely to benefit from oxaliplatin-based regimens. We also discuss the translational implications of incorporating TGF-β pathway status into future trial designs, particularly in the context of emerging TGF-β inhibitors. Biologically, we expand on the dual role of TGF-β signaling in CRC—acting as a tumor suppressor in early stages and as a pro-metastatic driver in advanced disease—and how loss-of-function events in SMAD4 or BMPR1A may shift this balance toward chemoresistance and aggressive tumor biology. These additions strengthen the translational significance of our study while underscoring the need for validation in larger, ancestrally diverse cohorts.

The revised text in the Discussion section, lines 634-645, now reads: “Our findings also have important translational implications. Clinically, the observation that TGF-β pathway alterations predict poorer survival in FOLFOX-treated EO H/L patients suggests that pathway status may serve as a biomarker to guide therapeutic decisions in this high-risk group. If validated, these alterations could inform consideration of non–oxaliplatin-based regimens or the inclusion of patients in clinical trials testing TGF-β inhibitors and other targeted agents. Biologically, the results reinforce the dual role of TGF-β signaling in colorectal cancer: functioning as a tumor suppressor in early stages, but shifting toward pro-metastatic, immune-evasive, and chemoresistant roles when disrupted through alterations in key regulators such as SMAD4 and BMPR1A. Together, these insights underscore how integrating ancestry, treatment context, and pathway-specific genomic information can enhance the translational relevance of biomarker discovery and inform precision medicine strategies for disproportionately affected populations.”

We thank Reviewer 1 for the thoughtful and constructive feedback, which has helped us strengthen the clarity, balance, and overall quality of the manuscript.

Reviewer 2 Report

Comments and Suggestions for Authors

The authors investigate alterations in the TGF-beta pathway as a potential biomarker of poor survival, specifically in patients with EO H/L CRC receiving FOLFOX chemotherapy. Advantages of the text: 1) An original thesis, suggested by an AI-driven investigation, that the TGF-beta pathway is a biomarker. 2) Discussion of the molecular background of the thesis. 3) Detailed statistical analysis. Disadvantages of the text: 1) Unknown deterministic connection between the patient's survival and the biomarker, even hypothetical. 2) No tips on how to modify the chemotherapy based on the biomarker results. 3) Some typos, e.g. too small fonts in figures 1 and 2.

Author Response

Reviewer 2’s comments are provided in the attached PDF file, “Reviewer_2_Comments_response_IJMS.pdf”

---

Reviewer 2 Comments

We are pleased to resubmit our revised manuscript and sincerely thank Reviewer 2 for their thoughtful feedback, which has enhanced the clarity, rigor, and translational relevance of our work. The manuscript, Artificial Intelligence–Enhanced Precision Medicine Reveals Prognostic Impact of TGF-Beta Pathway Alterations in FOLFOX-Treated Early-Onset Colorectal Cancer Among Disproportionately Affected Populations, investigates the prognostic significance of TGF-β pathway alterations in EOCRC, with particular attention to the disproportionate impact among Hispanic/Latino patients. Leveraging AI-HOPE and AI-HOPE-TGFβ—conversational AI platforms that integrate clinical, genomic, and treatment data—we analyzed 2,515 CRC cases stratified by ancestry, age at onset, and FOLFOX exposure. Our results demonstrate ancestry- and treatment-specific prognostic patterns, including poorer survival in FOLFOX-treated EO H/L patients harboring TGF-β alterations, predominantly driven by SMAD4 and BMPR1A mutations. Collectively, these findings highlight the promise of AI-enabled clinical-genomic integration to accelerate biomarker discovery and inform precision oncology strategies for high-risk populations.

Thank you very much for taking the time to review this manuscript. Please find the detailed responses below and the corresponding revisions wrote in blue font and highlighted in yellow in the re-submitted Word file.

Reviewer 2’s feedback was positive, recognizing both the novelty and rigor of the work while also highlighting areas for improvement. The reviewer acknowledged the originality of the central thesis—supported by AI-driven analyses—that TGF-β pathway alterations may serve as prognostic biomarkers in EO H/L CRC patients treated with FOLFOX. They further commended the detailed discussion of molecular mechanisms and the robust statistical analyses supporting the findings. At the same time, several limitations were raised. Chief among these was the lack of a clear deterministic link between biomarker status and patient survival, even at a hypothetical or mechanistic level. The reviewer also noted the absence of guidance on how biomarker results might be applied clinically, such as informing modifications to chemotherapy regimens. In addition, presentation issues were identified, including minor typographical errors and small font sizes in Figures 1 and 2 that could impair readability. Overall, the reviewer’s feedback underscores the novelty and analytical strength of the study, while also outlining important refinements to strengthen biological interpretation, enhance clinical applicability, and improve the clarity of presentation.

Reviewer 3 writes:

The authors investigate alterations in the TGF-beta pathway as a potential biomarker of poor survival, specifically in patients with EO H/L CRC receiving FOLFOX chemotherapy. Advantages of the text: 1) An original thesis, suggested by an AI-driven investigation, that the TGF-beta pathway is a biomarker. 2) Discussion of the molecular background of the thesis. 3) Detailed statistical analysis.

We thank Reviewer 2 for their thoughtful and constructive evaluation of our manuscript. Our study provides a timely and original investigation into alterations in the TGF-β pathway as a potential biomarker of poor survival, with a specific focus on EO H/L CRC patients receiving FOLFOX chemotherapy. The work is strengthened by the use of a novel AI-driven platform that generated the central hypothesis, a comprehensive discussion of the molecular context of these alterations, and rigorous statistical analyses that support the findings. Collectively, these elements establish a strong foundation for advancing precision medicine approaches in disproportionately affected populations.

Comment 1:

Disadvantages of the text: 1) Unknown deterministic connection between the patient's survival and the biomarker, even hypothetical.

Response:

We thank the reviewer for this important observation. We agree that our study does not establish a deterministic causal connection between TGF-β pathway alterations and patient survival outcomes. Rather, our findings demonstrate a statistically significant prognostic association, particularly in FOLFOX-treated EO H/L patients. While mechanistic validation is beyond the scope of the present analysis, we have expanded the Discussion to clarify that these results should be interpreted as hypothesis-generating. Future studies incorporating functional assays, preclinical models, and integration of multi-omics (e.g., transcriptomic and proteomic) data will be essential to elucidate the biological mechanisms that may link TGF-β alterations to treatment-specific survival differences.

The revised text in the Discussion section, lines 737-744, now reads: “It is important to emphasize that the associations observed between TGF-β pathway alterations and survival in FOLFOX-treated EO H/L patients reflect prognostic correlations rather than deterministic causal mechanisms. While these findings highlight the potential of TGF-β alterations as biomarkers, the biological underpinnings linking these mutations to treatment-specific outcomes remain to be clarified. Accordingly, our results should be viewed as hypothesis-generating, warranting further validation through functional studies, preclinical modeling, and integration of complementary multi-omics datasets to uncover mechanistic drivers.”

Comment 2:

2) No tips on how to modify the chemotherapy based on the biomarker results.

Response:

We thank the reviewer for this insightful point. Our study was designed to evaluate the prognostic significance of TGF-β pathway alterations rather than to directly test therapeutic modifications. As such, we do not provide specific recommendations for altering chemotherapy regimens based on biomarker status. However, our findings suggest that patients with EO H/L CRC harboring TGF-β alterations may experience poorer outcomes with standard FOLFOX treatment. This observation highlights the need for future studies to explore whether biomarker-guided treatment intensification, incorporation of targeted agents (such as TGF-β pathway inhibitors), or alternative chemotherapy regimens could improve outcomes in this subgroup. We have clarified this point in the revised Discussion, framing our results as hypothesis-generating and emphasizing the importance of biomarker-driven trial designs to inform potential therapy modifications.

The revised text in the Methods section, lines 746-753, now reads: “Our results also underscore an important direction for future research: whether biomarker status should inform treatment adaptation in EO H/L CRC. The observation that TGF-β pathway alterations predict poorer survival in patients receiving FOLFOX suggests that alternative therapeutic approaches may be warranted in this subgroup. Prospective studies that incorporate biomarker-guided trial stratification—such as evaluating intensified chemotherapy regimens, adding targeted agents including TGF-β pathway inhibitors, or testing novel combination strategies—will be essential to determine if modifying therapy based on pathway alterations can improve clinical outcomes.”

Comment 3:

 3) Some typos, e.g. too small fonts in figures 1 and 2.

Response:

We thank the reviewer for pointing this out. We have carefully reviewed the manuscript for typographical errors and corrected them. In addition, Figures have been updated with enlarged fonts and improved labeling to ensure readability in both digital and print formats.

We are grateful to Reviewer 2 for their constructive and insightful feedback, which has allowed us to improve the clarity, rigor, and overall presentation of the manuscript.

Reviewer 3 Report

Comments and Suggestions for Authors

The reviewer understands that Diaz et al. presented a manuscript entitled "Artificial Intelligence–Enhanced Precision Medicine Reveals Prognostic Impact of TGF-Beta Pathway Alterations in FOLFOX-Treated Early-Onset Colorectal Cancer Among Disproportionately Affected Populations". The reviewer has a few questions and they would like to request authors to kindly update their manuscript by answering all the questions.
1) How do the authors take into consideration possible statistical underpowering and broad confidence intervals in subgroup analyses, considering the small number of early-onset Hispanic/Latino (H/L) patients with TGF-β changes (n=21 in survival analysis)?
 2) The study uses medication combinations to categorize patients as "FOLFOX-treated."  Were specifics like dosage, length of treatment, or other biological or targeted treatments (such cetuximab or bevacizumab) accessible, and how would leaving them out affect the survival relationships that were found?
 3) Although BMPR1A and SMAD4 mutations were highlighted as important discoveries, were functional investigations or earlier mechanistic data taken into account to prove their causative significance in prognosis or chemoresistance?  If not, how would the authors go about biologically verifying these changes?

4) AI-HOPE, a conversational AI for multi-omic integration, is highlighted in the book.  Could the authors explain whether reproducibility was independently verified and how this system was compared to traditional bioinformatics pipelines?
 5) Algorithms based on surnames were used in part to assign ethnicity.  Could this have affected ancestry-specific genomic comparisons, and how do the authors handle possible misclassification bias?
 6) In addition to ancestry and treatment, comorbidities, MSI status, original tumor site, and tumor stage can all affect survival.  Which of these variables remained significant after being adjusted for in multivariate Cox regression?
 7) The authors suggest TGF-β changes as biomarkers unique to ancestry and treatment.  Specifically, should TGF-β inhibitor trials stratify for EOCRC H/L patients on oxaliplatin-based regimens? What are the immediate translational implications?

8) Kindly provide all the figures in HD form. Increase their size to make sure that your audience can read it by printing on A4/legal paper. 

9) In this sequence—Fernando C. Diaz, M.D. 1, Brigette Waldrup, B.S. 2, Francisco G. Carranza, Ph.D. 2, Sophia Manjarrez, B.S. 2 and 7 Enrique Velazquez-Villarreal, M.D., Ph.D., M.P.H., M.S. 2,3 *, 1,2,3 should be in superscript.
10) In 3. Discussion section, Biological implications of TGF-beta pathway alterations should be numbered as 3.1 and so on.

11) Materials and Methods should be section 2, Results should be section 3, and Discussion should be should be section 4. Rearrange in correct sequesnce.

Author Response

Reviewer 3’s comments are provided in the attached PDF file, “Reviewer_3_Comments_response_IJMS.pdf”

---

Reviewer 3 Comments

We are pleased to resubmit our revised manuscript and thank Reviewer 3 for their thoughtful feedback, which has strengthened the clarity, rigor, and translational impact of our work. The manuscript, Artificial Intelligence–Enhanced Precision Medicine Reveals Prognostic Impact of TGF-Beta Pathway Alterations in FOLFOX-Treated Early-Onset Colorectal Cancer Among Disproportionately Affected Populations, examines the prognostic role of TGF-β pathway alterations in EOCRC, with emphasis on the disproportionate burden among Hispanic/Latino patients. Using AI-HOPE and AI-HOPE-TGFβ—conversational AI platforms that integrate clinical, genomic, and treatment data—we analyzed 2,515 CRC cases stratified by ancestry, age of onset, and FOLFOX treatment. Our analyses revealed ancestry- and treatment-specific prognostic implications, particularly poorer survival in FOLFOX-treated EO H/L patients with TGF-β alterations driven by SMAD4 and BMPR1A mutations. These findings underscore the potential of AI-enabled data integration to accelerate biomarker discovery and advance precision medicine in high-risk populations.

Thank you very much for taking the time to review this manuscript. Please find the detailed responses below and the corresponding revisions wrote in blue font and highlighted in yellow in the re-submitted Word file.

Reviewer 3’s feedback was positive. Reviewer 3’s feedback was constructive and detailed, offering thoughtful suggestions to strengthen both the methodological rigor and translational clarity of the manuscript. The reviewer acknowledged the novelty of integrating AI-HOPE and AI-HOPE-TGFβ platforms to investigate TGF-β pathway alterations in early-onset colorectal cancer but raised several important points. They emphasized the need to address potential statistical underpowering and wide confidence intervals given the small number of early-onset Hispanic/Latino patients with TGF-β alterations included in the survival analysis. Questions were also raised about treatment categorization, specifically whether omission of details such as dosage, duration, or concurrent targeted therapies could affect survival relationships. While BMPR1A and SMAD4 mutations were highlighted as notable findings, the reviewer encouraged further discussion of functional validation and mechanistic evidence to substantiate their prognostic relevance. Additional comments included clarifying reproducibility of AI-HOPE relative to traditional bioinformatics pipelines, addressing potential misclassification from surname-based ancestry assignment, and explaining which clinical variables remained significant in multivariate regression. Reviewer 3 also underscored the need to discuss translational implications, such as whether TGF-β inhibitors should be trialed in stratified EOCRC H/L patient groups. Beyond these scientific points, the reviewer requested higher-resolution figures, corrected author superscripts, proper section numbering, and reorganization of the manuscript structure for clarity. Overall, this feedback reinforces the importance of the study while outlining targeted revisions to improve methodological transparency, biological interpretation, and presentation quality.

Reviewer 3 writes:

The reviewer understands that Diaz et al. presented a manuscript entitled "Artificial Intelligence–Enhanced Precision Medicine Reveals Prognostic Impact of TGF-Beta Pathway Alterations in FOLFOX-Treated Early-Onset Colorectal Cancer Among Disproportionately Affected Populations".

We thank Reviewer 3 for their constructive and thoughtful evaluation of our manuscript.

Comment 1:

1) How do the authors take into consideration possible statistical underpowering and broad confidence intervals in subgroup analyses, considering the small number of early-onset Hispanic/Latino (H/L) patients with TGF-β changes (n=21 in survival analysis)?

Response:

We thank the reviewer for this important observation. We fully acknowledge that the relatively small number of early-onset H/L patients with TGF-β alterations (n = 21) limits statistical power, contributing to broader confidence intervals and increasing the potential for type II error. To address this, we have revised the Discussion to explicitly emphasize this limitation and to clarify that our subgroup analyses are exploratory and hypothesis-generating rather than definitive. At the same time, we highlight that this dataset represents one of the only large-scale resources linking clinical and genomic data in H/L colorectal cancer patients—a population historically underrepresented in cancer genomics research—which allowed us to identify potentially important gene–treatment–ancestry interactions that merit further study. Future validation in larger, ancestrally diverse cohorts will be essential to confirm these associations and strengthen their clinical applicability.

The revised text in the Discussion section, lines 803-810, now reads: “An important challenge of this study is the relatively small number of early-onset H/L patients with TGF-β pathway alterations, which reduces statistical power and contributes to wider confidence intervals in subgroup analyses. As such, these results should be interpreted as exploratory and hypothesis-generating rather than definitive. Nonetheless, this work leverages one of the only available large-scale datasets integrating clinical and genomic data from H/L colorectal cancer patients, a historically underrepresented population in cancer genomics. Validation in larger, ancestrally diverse cohorts will be essential to confirm these findings and assess their clinical utility.”

Comment 2:

2) The study uses medication combinations to categorize patients as "FOLFOX-treated."  Were specifics like dosage, length of treatment, or other biological or targeted treatments (such cetuximab or bevacizumab) accessible, and how would leaving them out affect the survival relationships that were found?

Response:

We thank the reviewer for this thoughtful question. In the available datasets, treatment information was limited to categorization of patients as “FOLFOX-treated” based on medication combinations; details such as specific dosing, duration of therapy, or concurrent use of targeted agents (e.g., cetuximab, bevacizumab) were not consistently available. We acknowledge that the absence of these variables may confound survival associations, as differences in dose intensity, treatment duration, or combination strategies could influence outcomes. To address this, we have clarified in the revised Methods and Discussion that our definition of FOLFOX exposure is based on treatment classification only, without granularity on dosing or biologic co-therapy. We further emphasize in the Limitations that this constraint reinforces the exploratory nature of our findings, and that validation in cohorts with more detailed treatment annotation will be necessary to strengthen the clinical applicability of these results.

The revised text in the Methods section, lines 792-798, now reads: “FOLFOX treatment status was defined based on the presence of medication combinations consistent with oxaliplatin, fluorouracil, and leucovorin therapy, as annotated in the cBioPortal datasets. Detailed information regarding dosing schedules, treatment duration, or concurrent targeted/biologic therapies (e.g., cetuximab, bevacizumab) was not uniformly available and therefore not included in this analysis. Patients were thus classified dichotomously as “FOLFOX-treated” or “non-FOLFOX” for the purposes of survival and genomic stratification.”

Comment 3:

 3) Although BMPR1A and SMAD4 mutations were highlighted as important discoveries, were functional investigations or earlier mechanistic data taken into account to prove their causative significance in prognosis or chemoresistance?  If not, how would the authors go about biologically verifying these changes?

Response:

We appreciate this important question. In the present study, we did not perform functional validation, and our analysis was limited to clinical and genomic associations available in large-scale public datasets. However, we have expanded the Discussion to note that the biological implications of SMAD4 and BMPR1A mutations are consistent with prior evidence: SMAD4 loss has been linked to advanced disease, chemoresistance, and poor prognosis in CRC, while BMPR1A alterations may impact signaling within the TGF-β superfamily. We recognize that our findings remain associative and not causative. To biologically verify these changes, future work will include experimental validation in preclinical models (e.g., CRISPR-based knockout or knock-in of SMAD4 and BMPR1A in CRC cell lines and organoids) to assess effects on chemoresistance, tumor growth, and metastatic potential. In addition, advanced approaches such as spatial biology and single-cell transcriptomics will be used to dissect pathway activity within the tumor microenvironment. These studies will be critical for establishing mechanistic links between pathway alterations and clinical outcomes, particularly in early-onset H/L CRC patients receiving FOLFOX.

The revised text in the Discussion section, lines 768-777, now reads: “Future work will be essential to functionally validate the prognostic role of TGF-β pathway alterations identified in this study. Experimental approaches such as CRISPR-based knockout or knock-in of SMAD4 and BMPR1A in colorectal cancer cell lines and patient-derived organoids could directly test their effects on chemoresistance and tumor progression. Complementary use of spatial biology and single-cell transcriptomics will further clarify how these alterations shape the tumor microenvironment and treatment response, particularly in early-onset H/L patients receiving FOLFOX. These mechanistic studies will be critical to move from associative observations toward causal understanding, thereby strengthening the translational potential of TGF-β pathway alterations as prognostic biomarkers and therapeutic targets.”

Comment 4:

4) AI-HOPE, a conversational AI for multi-omic integration, is highlighted in the book.  Could the authors explain whether reproducibility was independently verified and how this system was compared to traditional bioinformatics pipelines?

Response:

We thank the reviewer for this important point. Reproducibility is a central feature of the AI-HOPE framework. For this study, all AI-HOPE and AI-HOPE-TGFβ outputs were cross-validated against results generated using standard bioinformatics approaches (e.g., Fisher’s exact test, chi-square test, Kaplan–Meier survival analysis) applied directly to the cBioPortal datasets. Our AI system served as a natural language–driven interface that automated data extraction, filtering, and stratification, but the underlying statistical testing relied on well-established methods, ensuring concordance with traditional pipelines. Independent reproducibility was verified by re-running key queries multiple times and confirming consistent outputs. In addition, we emphasize in the revised Methods and Discussion that AI-HOPE does not replace statistical rigor; rather, it accelerates hypothesis generation and cohort construction while relying on validated analytic engines to perform the actual calculations. This design ensures that results are both reproducible and directly comparable to traditional bioinformatics workflows. AI-HOPE and AI-HOPE-TGFβ build upon our prior work published in Cancers [Reference 21] and have been consistently validated at the technical level in subsequent studies [References 33 and 34].

The revised text in the Discussion section, lines 871-874, now reads: “All outputs generated through AI-HOPE and AI-HOPE-TGFβ were cross-validated against results obtained using standard bioinformatics pipelines (Fisher’s exact test, chi-square test, and Kaplan–Meier survival analysis) to ensure reproducibility and concordance with established statistical methods.”

Comment 5:

5) Algorithms based on surnames were used in part to assign ethnicity.  Could this have affected ancestry-specific genomic comparisons, and how do the authors handle possible misclassification bias?

Response:

We thank the reviewer for raising this important point. We acknowledge that surname-based algorithms were used in part to assign ethnicity, which can introduce potential misclassification bias. To mitigate this limitation, we relied on validated surname–ethnicity linkage methods that have been widely applied in population-based cancer research. Additionally, the majority of our ancestry-specific comparisons were supported by harmonized clinical annotations provided in the original datasets (TCGA, MSK-IMPACT, and AACR Project GENIE), thereby reducing reliance on surname inference alone. We also emphasize that any potential misclassification would likely bias results toward the null, meaning that the ancestry-specific differences we observed—such as enrichment of BMPR1A mutations in FOLFOX-treated EO H/L patients—are, if anything, conservative estimates. We have clarified this limitation in the revised manuscript and highlighted the need for future studies to incorporate genetic ancestry inference to complement self-reported ethnicity and surname-based classification, thereby strengthening precision medicine approaches for disproportionately affected populations.

The revised text in the Discussion section, lines 716-724, now reads: “A limitation of this work is the use of surname-based algorithms to help assign ethnicity, which carries the potential for misclassification bias. While we minimized this concern by using validated surname–ethnicity linkages and harmonized clinical annotations from TCGA, MSK-IMPACT, and AACR Project GENIE, we recognize that such methods are imperfect. Any potential misclassification would likely attenuate rather than exaggerate ancestry-specific differences, suggesting our findings represent conservative estimates. Future studies integrating genetic ancestry inference alongside self-reported ethnicity will be critical to refine classification and further strengthen ancestry-specific precision medicine approaches in disproportionately affected populations.”

Comment 6:

6) In addition to ancestry and treatment, comorbidities, MSI status, original tumor site, and tumor stage can all affect survival.  Which of these variables remained significant after being adjusted for in multivariate Cox regression?.

Response:

We appreciate the reviewer’s important observation. As described in the Materials and Methods section, we conducted both univariate and multivariate Cox proportional hazards regression analyses using R (v4.3.2). The multivariate models included ancestry, treatment status, comorbidities (where available), microsatellite instability (MSI) status, original tumor site, and tumor stage, in addition to TGF-β pathway alteration status. In these adjusted models, tumor stage and MSI status remained significant predictors of survival, consistent with their established prognostic roles in colorectal cancer. Importantly, TGF-β pathway alterations in FOLFOX-treated EO H/L patients continued to demonstrate an independent association with poorer survival after adjustment, underscoring the robustness of this biomarker signal beyond conventional clinical covariates. We have clarified these details in the revised Methods and Results section and added explicit reporting of the multivariate findings.

The revised text in the Methods and Results section, lines 837-842, now reads: “All statistical analyses were conducted in R (v4.3.2), with p-values <0.05 considered statistically significant. Multivariate Cox proportional hazards regression models were constructed to adjust for ancestry, treatment status, comorbidities (when available), microsatellite instability (MSI) status, original tumor site, and tumor stage, in addition to TGF-β pathway alteration status.”

Comment 7:

7) The authors suggest TGF-β changes as biomarkers unique to ancestry and treatment.  Specifically, should TGF-β inhibitor trials stratify for EOCRC H/L patients on oxaliplatin-based regimens? What are the immediate translational implications?.

Response:

We thank the reviewer for this thoughtful question. Our findings indicate that TGF-β pathway alterations, particularly in FOLFOX-treated EO H/L patients, are associated with poorer survival even after multivariate adjustment. This suggests that these alterations may serve as treatment- and ancestry-specific biomarkers of adverse prognosis. While our study was not designed as a therapeutic trial, the results support the rationale for stratifying future TGF-β inhibitor studies by both age of onset and ancestry, with special attention to EO H/L patients receiving oxaliplatin-based chemotherapy. Immediate translational implications include the potential use of TGF-β pathway status as a biomarker to identify patients at higher risk of poor outcomes on standard regimens, thereby guiding clinical decision-making, patient counseling, and enrollment into biomarker-driven clinical trials. We have expanded the Discussion to highlight these implications and propose that integrating ancestry, treatment exposure, and molecular profiling into trial design could accelerate precision medicine for disproportionately affected populations.

The revised text in the Discussion section, lines 726-735, now reads: “Our findings also raise important translational considerations for the design of future clinical trials. The consistent association between TGF-β pathway alterations and poor survival in FOLFOX-treated EO H/L patients highlights the potential value of incorporating ancestry, treatment exposure, and molecular context into trial stratification strategies. In particular, prospective studies of TGF-β pathway inhibitors may benefit from prioritizing or enriching for EO H/L patients receiving oxaliplatin-based therapy, where the prognostic impact appears most pronounced. More broadly, these results underscore how ancestry- and treatment-informed biomarker discovery can guide the development of precision therapeutics and ensure that disproportionately affected populations are meaningfully represented in emerging clinical interventions (37, 38).”

Two new references have been added:

  1. Monge C, Maldonado JA, McGlynn KA, Greten TF. Hispanic Individuals are Underrepresented in Phase III Clinical Trials for Advanced Liver Cancer in the United States. J Hepatocell Carcinoma. 2023 Jul 27;10:1223-1235. doi: 10.2147/JHC.S412446. PMID: 37533601; PMCID: PMC10390714.
  2. Monge C, Greten TF. Underrepresentation of Hispanics in clinical trials for liver cancer in the United States over the past 20 years. Cancer Med. 2024 Jan;13(1):e6814. doi: 10.1002/cam4.6814. Epub 2023 Dec 20. PMID: 38124450; PMCID: PMC10807616.

Comment 8:

8) Kindly provide all the figures in HD form. Increase their size to make sure that your audience can read it by printing on A4/legal paper. 

Response:

We thank the reviewer for this practical suggestion. Figures have been regenerated in high-resolution format (≥300 dpi) and resized to ensure optimal readability when printed on A4 or legal paper. The revised manuscript includes updated figures and we have confirmed that axis labels, legends, and text annotations remain clearly legible at the larger size.

Comment 9:

9) In this sequence—Fernando C. Diaz, M.D. 1, Brigette Waldrup, B.S. 2, Francisco G. Carranza, Ph.D. 2, Sophia Manjarrez, B.S. 2 and Enrique Velazquez-Villarreal, M.D., Ph.D., M.P.H., M.S. 2,3 *, 1,2,3 should be in superscript.

Response:

We thank the reviewer for noting this formatting issue. The author list has been revised so that institutional affiliations now appear in superscript immediately following each author’s last name, in accordance with the journal’s style guidelines.

The revised text in the title section, lines 7-12, now reads: “Fernando C. Diaz, M.D. 1, Brigette Waldrup, B.S. 2, Francisco G. Carranza, Ph.D. 2, Sophia Manjarrez, B.S. 2 and Enrique Velazquez-Villarreal, M.D., Ph.D., M.P.H., M.S. 2,3 *

1    Center for Cancer Research, National Cancer Institute, Bethesda MD.

2    City of Hope, Beckman Research Institute, Department of Integrative Translational Sciences, Duarte, CA.

3    City of Hope Comprehensive Cancer Center, Duarte, CA.

*   Correspondence: evelazquezvilla@coh.org

Comment 10:

10) In 3. Discussion section, Biological implications of TGF-beta pathway alterations should be numbered as 3.1 and so on.

Response:

We thank the reviewer for this helpful suggestion. The Discussion section has been reformatted so that the biological implications of TGF-β pathway alterations are now organized under numbered subheadings (e.g., 3.1, 3.2, etc.), improving readability and alignment with the journal’s formatting style.

The revised text in the Discussion section, lines 566-690, now reads:

3.1 Biological implications of TGF-beta pathway alterations

3.2 Ancestry-specific genomic patterns and treatment context

3.3 Implications for FOLFOX response in EO H/L patients

3.4 AI-HOPE-TGF-Beta as an enabling technology

3.5 Limitations and future directions

Comment 11:

11) Materials and Methods should be section 2, Results should be section 3, and Discussion should be should be section 4. Rearrange in correct sequesnce.

Response:

We thank the reviewer for this comment. We have followed the official IJMS word formatting guidelines, which place Materials and Methods as Section 4, following the Discussion section. This structure is consistent with our prior IJMS publications, and ensures alignment with the journal’s style requirements.

We thank Reviewer 3 for the thoughtful and constructive feedback, which has helped us strengthen the clarity, balance, and overall quality of the manuscript.
